# Leukemic stem cells activate lineage inappropriate signalling pathways to promote their growth

Sophie G. Kellaway ®[1,6] ✉, Sandeep Potluri ®[1], Peter Keane ®[1,7], Helen J. Blair[2], Luke Ames[1], Alice Worker[1], Paulynn S. Chin ®[1], Anetta Ptasinska[1], Polina K. Derevyanko ®[3], Assunta Adamo[1], Daniel J. L. Coleman ®[1], Naeem Khan[4], Salam A. Assi[1], Anja Krippner-Heidenreich[3], Manoj Raghavan[1,5], Peter N. Cockerill ®[1], Olaf Heidenreich ®[2,3,8] & Constanze Bonifer ®[1,8] ✉

Acute Myeloid Leukemia (AML) is caused by multiple mutations which dys-regulate growth and differentiation of myeloid cells. Cells adopt different gene regulatory networks specific to individual mutations, maintaining a rapidly proliferating blast cell population with fatal consequences for the patient if not treated. The most common treatment option is still chemotherapy which targets such cells. However, patients harbour a population of quiescent leu-kemic stem cells (LSCs) which can emerge from quiescence to trigger relapse after therapy. The processes that allow such cells to re-grow remain unknown. Here, we examine the well characterised t(8;21) AML sub-type as a model to address this question. Using four primary AML samples and a novel t(8;21) patient-derived xenograft model, we show that t(8;21) LSCs aberrantly activate the VEGF and IL-5 signalling pathways. Both pathways operate within a reg-ulatory circuit consisting of the driver oncoprotein RUNX1::ETO and an AP-1/GATA2 axis allowing LSCs to re-enter the cell cycle while preserving self-renewal capacity.

Acute myeloid leukemia (AML) is a disease characterized by excessive production of leukemic blast cells with impaired differentiation capacity. This blast population is replenished by rare leukemia initi-ating cells, called leukemic stem cells (LSCs)[1–3]. Similar to healthy hematopoietic stem cells (HSCs), LSCs are generally quiescent[2] and are therefore thought to be responsible for relapse following che-motherapy which targets rapidly proliferating cells. Thus, relapse depends on signals that induce LSCs to re-enter the cell cycle, to proliferate to generate blasts and to repopulate the AML[4]. Whether LSCs are quiescent or proliferating is likely to be the result of

transcriptional control in cooperation with signalling processes oper-ating in the niche occupied by the cells. LSCs utilise similar, but not identical, growth control mechanisms as compared to HSCs[4], which may allow for selective targeting. For example, FoxM1 regulates the cell cycle specifically in MLL-rearranged LSCs[5,6]. Different subtypes of AML are caused by different mutations, and we have shown that blast cells adopt subtype-specific gene regulatory networks (GRNs) main-taining the leukemic phenotype[7,8]. It is largely unknown how GRNs in LSCs compare to those of blast cells as the latter dominate the tran-scriptional signature in bulk sequencing analysis.

[1]Institute of Cancer and Genomic Sciences, University of Birmingham, Birmingham, UK. [2]Translational and Clinical Research Institute, Newcastle University, Newcastle upon Tyne, UK. [3]Princess Maxima Center of Pediatric Oncology, Utrecht, Netherlands. [4]Institute of Immunology and Immunotherapy, University of Birmingham, Birmingham, UK. [5]Centre for Clinical Haematology, Queen Elizabeth Hospital, Birmingham, UK. [6]Present address: Blood Cancer and Stem Cells, Centre for Cancer Sciences, School of Medicine, University of Nottingham, Nottingham, UK. [7]Present address: School of Biosciences, University of Bir-mingham, Birmingham, UK. [8]These authors jointly supervised this work: Olaf Heidenreich, Constanze Bonifer. ✉e-mail: Sophie.Kellaway@nottingham.ac.uk; c.bonifer@bham.ac.uk

AML driven by the t(8;21) chromosomal translocation is one of the best characterised common subtypes. Remission is achieved in around 90% of t(8;21) patients but they are prone to relapse associated with poor outcomes[9]. The translocation produces the RUNX1::ETO fusion protein, and the resulting AML displays a unique GRN[7,10,11]. The RUNX1::ETO oncogene is expressed under the control of the *RUNX1* promoters and the protein interferes with the normal action of RUNX1 by binding to the same sites in the genome[12,13]. Furthermore, both cell extrinsic and intrinsic signalling are known to play roles in t(8;21) growth, whereby activation of the AP-1 pathway upregulates transcription of signalling and cell cycle genes[14–16]. t(8;21) AML is therefore an attractive model to study LSC activation and to identify targets aimed at preventing relapse. In this study, we determined the genome-wide t(8;21) LSC-specific open chromatin structure and gene expression profile. We also profiled LSCs at the single cell level using single cell RNA-Seq (scRNA-Seq) which, together with perturbation experiments using a novel patient derived xenograft (PDX) model for t(8;21) AML, identified the growth factors VEGFA and IL-5 and their receptors as key factors aberrantly driving the growth of this specific LSC subtype. Furthermore, we identify an oncoprotein driven transcription/ signalling circuit, dependent on the AP-1 family of transcription factors as mediators of VEGF/IL-5 signalling that regulates the balance between LSC maintenance and blast growth.

## Results

### t(8;21) LSCs exhibit mutation-specific gene expression and chromatin accessibility profiles

We employed t(8;21) AML as an archetypal model system to gain an understanding of the factors which activate LSC growth and drive relapse following chemotherapy. To examine whether LSC growth is controlled by a mutation subtype-specific or global mechanism, we defined transcriptional signatures for LSCs and blasts purified from four t(8;21) bone marrow/peripheral blood samples (referred to from hereon as t(8;21) #1-#4). Mutation profiling revealed that sample #1 carried mutations in several key hematopoietic regulator genes including *GATA1*, *KIT* and *WT1* at allele frequencies of 40% and higher, while sample #2 carried two FLT3 internal tandem duplications (ITD) and a *RAD21* indel at allele frequencies ≤10%, no additional mutations were identified in #3 and #4 (Table 1). Cells were sorted for the CD34+/ CD38− surface marker pattern to enrich for LSCs, with the CD34+/ CD38+ fraction comprising leukemic blasts (Supplementary Fig. 1A)[2,17], followed by genome-wide profiling of gene expression and open chromatin regions (Fig. 1A). Colony forming assays to verify the sorted populations produced 4 colonies per 1000 cells from sorted LSCs but zero colonies from blasts from t(8;21) #2 which carries a FLT3-ITD

positive subclone (Supplementary Fig. 1B). qRT-PCR confirmed that these colonies expressed RUNX1::ETO (Supplementary Fig. 1C). As is well known from t(8;21) AML[18], cells from the other patients produced no colonies.

In our previous studies we used DNaseI-seq to show that t(8;21) AML adopts a reproducible subtype-specific chromatin accessibility pattern[7]. We compared these DNaseI-seq data from bulk CD34+ AML cells with ATAC-seq data derived from CD34/CD38-sorted LSCs and blasts from t(8;21) #1 and #2 (based on available numbers of cells), and with healthy CD34+ peripheral blood stem cells (PBSCs). This analysis revealed that the t(8;21)-specific open chromatin signature, as compared to healthy cells, was also found in LSCs (Fig. 1B). However, when compared directly to blasts and ranked by the fold-change of tag counts per peak, the LSC chromatin accessibility profile differed from that of blast cells (Supplementary Fig. 1D). LSC-specific open chromatin sites were enriched for GATA binding motifs (Supplementary Fig. 1D, E) whilst the blast-specific sites were enriched for C/EBP and PU.1 motifs, indicating an accessibility pattern characteristic for more immature cells in LSCs[19,20].

To examine the heterogeneity of the sorted LSC population and to identify blast/LSC-specific gene signatures, we performed scRNA-seq on cells sorted as described above. Prior to sequencing, purified LSCs were enriched such that they were in an equal proportion with the blasts to better capture this rare population of interest. Cells were assigned in silico as LSC or blasts in the individual patients, before all four patient datasets were integrated (Fig. 1C) to reduce patient specific variability present in the merged dataset (Supplementary Fig. 1F). LSCs, blasts and a transition population were then assigned in the integrated dataset based on the contributing cells (Fig. 1C). LSC and blast specific marker genes were defined and confirmed in the individual patients (Supplementary Data 1, Supplementary Fig. 1G), and represent a t(8;21)-specific LSC signature. Blast-specific genes included *MPO* and neutrophil granule genes, *LYZ*, *AZU1* and *ELANE*. LSC-specific genes included transcription factors *KLF2*, *LMO4*, *SOX4*, *GATA2* and beta globin genes, the latter are known to be active in multipotent hematopoietic progenitor cells[20]. Trajectory analysis based on pseudotime placed *GATA2*-positive LSCs furthest from *MPO*-positive Blasts (Fig. 1D and Supplementary Fig. 1H).

Three clusters were identified in the LSC populations, with a further three clusters identified as representing the intermediate/transitional population (LSC/blast; Fig. 1E). Clusters showed good purity as defined by specific cluster markers (Supplementary Fig. 1I). We examined which clusters expressed t(8;21) AML-specific genes by defining 88 genes whose expression was at least 2-fold higher in t(8;21) patients compared to other AML subtypes or healthy CD34+ PBSCs

**Table 1 | Details of mutations in patient AML cells additional to the t(8;21) translocation, obtained from West Midlands Regional Genetics Laboratory**

| Patient | | Gene | Mutation | | VAF |
|---|---|---|---|---|---|
| t(8;21) #1 | Relapse | ETV6 | c.313_314insGG | NP_001978 p.R105fs | 45% |
| | | GATA1 | c.158 C > A | NP_002040 p.A53D | 51% |
| | | KIT | c.1253_1255del | NP_001087241 p.418_419del | 61% |
| | | NOTCH1 | c.4898 G > A | NP_060087 p.R1633H | 54% |
| | | NOTCH1 | c.6980 G > A | NP_060087 p.R2327Q | 48% |
| | | WT1 | c.420_421insGTGTGCGA | NP_001185481 p.R141fs | 39% |
| t(8;21) #2 | Presentation | RAD21 | c.1645delinsGGGGGTACT | NP_006256 p.Q549Gfs*66 | 10% |
| | | FLT3 | Internal Tandem Duplication | | 3% |
| | | FLT3 | Internal Tandem Duplication | | 1% |
| t(8;21) #3 | Relapse | Unknown | | | |
| t(8;21) #4 | Presentation | Unknown | | | |
| t(8;21) #5 | Relapse | KIT | c.2435 A > T | NP_000213.1 p.D816V | 48% |
| | | TET2 | c.4179delA | NP_001120680 p.T1393fs | 94% |

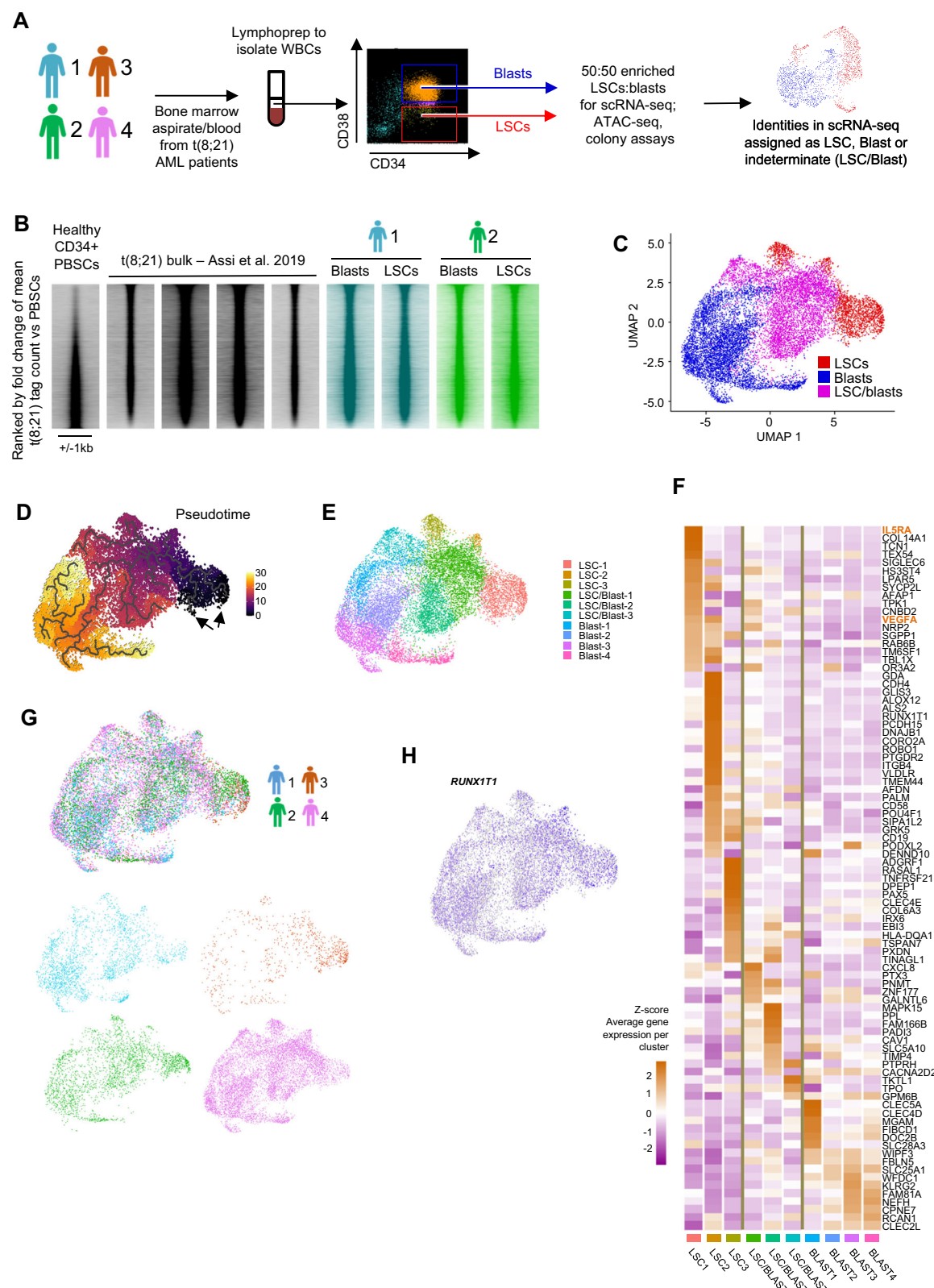

(Supplementary Fig. 1J), and plotting the Z-score of the expression of these 88 genes across the clusters ordered by pseudotime (Fig. 1F). The majority of these genes were most highly expressed in the LSC clusters, including genes important for the t(8;21) phenotype such as *POU4F1* and *PAX5*[21,22]. Cells from each patient were represented across all clusters (Fig. 1G) and expression of the *RUNX1T1* (ETO) transcript was detected in 6711/14485 cells (Fig. 1H) distributed across all clusters, thereby confirming that all cell clusters comprised AML or pre-

leukemic cells as *RUNX1T1* is not expressed in healthy myeloid cells[23]. Together these data show that the t(8;21)-specific gene regulatory network found in blasts is also found in LSCs.

## LSC and blast cells show cell cycle-specific gene expression heterogeneity

We next sought to identify genes regulating the growth status of LSCs and blast cells. We first assigned a cell cycle status to each cell using

**Fig. 1 | AML-subtype specific gene expression and chromatin accessibility is established in LSCs. A** Schematic showing how patient bone marrow cells were sorted into leukaemic stem cells (LSCs) and blasts for single cell RNA-seq, ATAC-seq (patients #1 and #2 only), and colony forming assays, gating strategy for sorting is shown in Supplementary Fig. 1A. **B** DNaseI-seq in t(8;21) patients and healthy CD34 + PBSCs[7] was ranked by the fold change of the average tag count in distal peaks and represented as density plots (+/−1 kb of the summit). ATAC-seq on sorted LSCs and blasts was plotted along the same axis. **C** UMAP plot of integrated scRNA-seq from four independent patients, where blue dots indicate cells assigned as blasts, red dots indicate cells assigned as LSCs and purple dots indicate

intermediate type cells which could not confidently be assigned as blasts or LSCs. **D** Pseudo-time trajectory analysis projected onto the UMAP plots, black arrows indicate the beginning of the trajectory. **E** Cell subclusters projected onto the UMAP plot. **F** Heatmaps with hierarchical clustering showing Z-scores of average gene expression per cluster of t(8;21) specific genes. **G** Cells from each individual patient projected onto the UMAP plot, together (top) and separately (bottom), with contributing cells coloured according to the patient of origin. Number of cells: t(8;21) #1 = 2489, #2 = 2546, #3 836 and #4 8664. **H** Expression of *RUNX1T1* projected onto the UMAP plot, where blue indicates the normalised UMI count.

scRNA-seq data which demonstrated a bias toward S-phase in blast cells and of $G_0/G_1$ in LSCs indicative of proliferation in blasts and quiescence in LSCs (Fig. 2A, 71% LSCs $G_0/G_1$ vs 43% blasts, 19% LSCs in S-phase vs 42% blasts). We also saw inter-patient cell cycle heterogeneity with more pronounced differences between LSC and blasts in t(8;21) #1 and #4 (Supplementary Fig. 2A). We then identified genes specifically expressed in cells from each cell cycle phase. Whilst S and $G_2/M$ phase LSCs and blasts both express genes essential for cell cycle regulation (Supplementary Data 2), $G_0/G_1$ overall gene expression was heterogeneous (Supplementary Fig. 2B). $G_0/G_1$ LSCs expressed genes associated with transcriptional control and negative regulation of the cell cycle, whilst the $G_0/G_1$ blast specific expression pattern was dominated by genes associated with translation and telomere maintenance, including elongation factors and ribosomal protein genes (Fig. 2B, Supplementary Data 2). The difference in expression of translation factors resembles the control of protein synthesis rates via phosphorylation of 4E-BP1, by which healthy HSCs regulate growth and quiescence[24]. In concordance with this result, the LSC-enriched GO-terms were generally more enriched in $G_0/G_1$ HSCs as compared to myeloid progenitors identified from published data[25] (Fig. 2B). The blast-specifically enriched GO-terms were not seen in healthy cells in line with the high levels of proliferation of AML cells, with the exception of $G_0/G_1$ myeloid progenitors showing enrichment for the translation GO-terms (Fig. 2B). To examine whether LSCs use a similar phosphorylation mechanism to HSCs, we performed mass cytometry on healthy PBSCs and cultured cells from patients #2 and #3 only, based on available numbers of cells. Phosphorylation of signalling molecules was overall higher in AML cells as compared to healthy hematopoietic cells (Fig. 2C). Proliferation, as determined by Ki67 was higher in CD34+/CD38+ blasts than in CD34+/CD38− LSCs in both patients and blasts contained increased levels of phosphorylated 4-EBP1 and S6 which are directly involved in control of protein translation (Fig. 2C and Supplementary Fig. 2C). Furthermore, phosphorylation of the AP-1 associated proteins CREB, JUN and JNK1/JNK2 was high in both patients, and more so in blasts than LSCs. In comparison, the STAT pathways and NF-κB were not differentially active. Together these data show that concordant with their quiescence, LSCs display reduced signalling activity influencing translation and the AP-1 pathway.

**Aberrant VEGF and IL-5 signalling in t(8;21) AML drives LSC activation and promotes the growth of a novel serially transplantable t(8;21) PDX model**

To identify candidate genes which could be involved in kick-starting LSC growth, we screened for cell signalling associated genes which were specifically expressed in t(8;21) LSCs. Signalling mutations such as in the *KIT* gene, giving rise to constitutively active receptor molecules, appeared to be insufficient to initiate LSC growth despite being equally present in LSCs and blasts. This was the case here, with t(8;21) #1 harbouring a *KIT* mutation detected in both blasts and quiescent LSCs. However, *VEGFA* and *IL5RA* mRNAs were found to be strongly enriched in LSCs and were largely t(8;21) specific (Fig. 1F, 2D, 2E). Furthermore, the gene encoding the VEGFA receptor (*KDR*) was also up-regulated in t(8;21) AML as compared to healthy PBSCs and all other

AML subtypes except for the CEBPAx2 subtype (Fig. 2E). In t(8;21) patients #2 and #4 we detected single *KDR*-expressing LSCs, albeit at a low level (Fig. 2D and Supplementary Fig. 2D) and in patient 1 *KDR* transcripts were detected in bulk RNA-seq data (raw FPKM in LSCs: 0.23, in blasts <0.01). *VEGFA* was also expressed in some blasts but was not generally co-expressed with *KDR* (Supplementary Fig. 2E, F). In concordance with the LSC-specific expression, *GATA2* showed a high degree of co-expression with *VEGFA*, *KDR* and *IL5RA* (Supplementary Fig. 2E, F). Furthermore, the VEGF co-receptor *NRP2* (Neuropilin)[26] was found to also be expressed in t(8;21) AML but with expression confined to the LSC/Blast transition cells. Expression in the transition cells may correspond to a mechanism by which VEGF stimulation initially occurs in the LSCs and is then further activated to stimulate growth (Supplementary Fig. 2G, H). Interestingly, mining our CHi-C data[7] showed that *KDR* and *KIT* share an enhancer[27] which displays greater accessibility in t(8;21) patient cells compared to PBSCs (Fig. 2F, upper panel). This result indicates a mutually exclusive regulation of *KIT* and *KDR* suggesting that the *KDR* expressing LSCs would not express *KIT* even in the event of a mutated *KIT* gene, which was confirmed by our scRNA-Seq data (Fig. 2F, lower panel), thus representing a potential additional mechanism controlling the aberrant activation of this gene.

To assess the roles of IL-5 and VEGF signalling we used two well-established t(8;21) cell line models: Kasumi-1 and SKNO-1[28,29]. IL-5 signalling could not be assessed in Kasumi-1 as this cell line does not express *IL5RA* nor displays accessible chromatin at this locus[28]. We cultured cell lines in the presence of exogenous VEGF or IL-5 and in all cases the growth rate increased after cytokine addition (Fig. 3A–C). We next used the VEGFA inhibitor bevacizumab[30], with no additional VEGF (as the AML cells express it already), and the IL5RA inhibitor benralizumab[31] with or without exogenous IL-5, to test whether the inhibitors would abrogate growth stimulation. Both inhibitors reduced growth rates and pushed cells into $G_0$ (Fig. 3A–D and Supplementary Fig. 3A–D). Note that not all cells express IL5RA or KDR on the surface (Supplementary Fig. 3E). The response to benralizumab was more pronounced with the addition of exogenous IL-5 (Fig. 3B and Supplementary Fig. 3B). Addition of IL-5 could not compensate for the dependency of SKNO-1 on GM-CSF, even though these cytokines signal via the same receptor beta chain (Supplementary Fig. 3F). Stimulation of growth by VEGF or IL-5 was specific to t(8;21) as growing non-t(8;21) AML cell lines in the presence of either growth factor had no effect (Supplementary Fig. 3G). Our data therefore show that VEGF and IL-5 signalling specifically promote the growth of t(8;21) AML cells.

t(8;21) cell lines are able to form colonies and so to evaluate the effect of inhibitors on this feature we carried out colony forming assays in the presence of the VEGF and IL-5 inhibitors. This experiment showed that bevacizumab led to a small reduction in the number of colonies formed initially in concordance with the reduced growth rate, whilst benralizumab had little impact upon primary colony forming capacity (Fig. 3E–G). However, when the colonies were replated, a significantly higher number of colonies were formed in the presence of bevacizumab or benralizumab+IL-5 relative to the controls, indicating a higher proportion of cells capable of self-renewing (Fig. 3E–G). Thus, blocking VEGF or IL-5 signalling stalls the cells in a less proliferative, but increased self-renewing state.

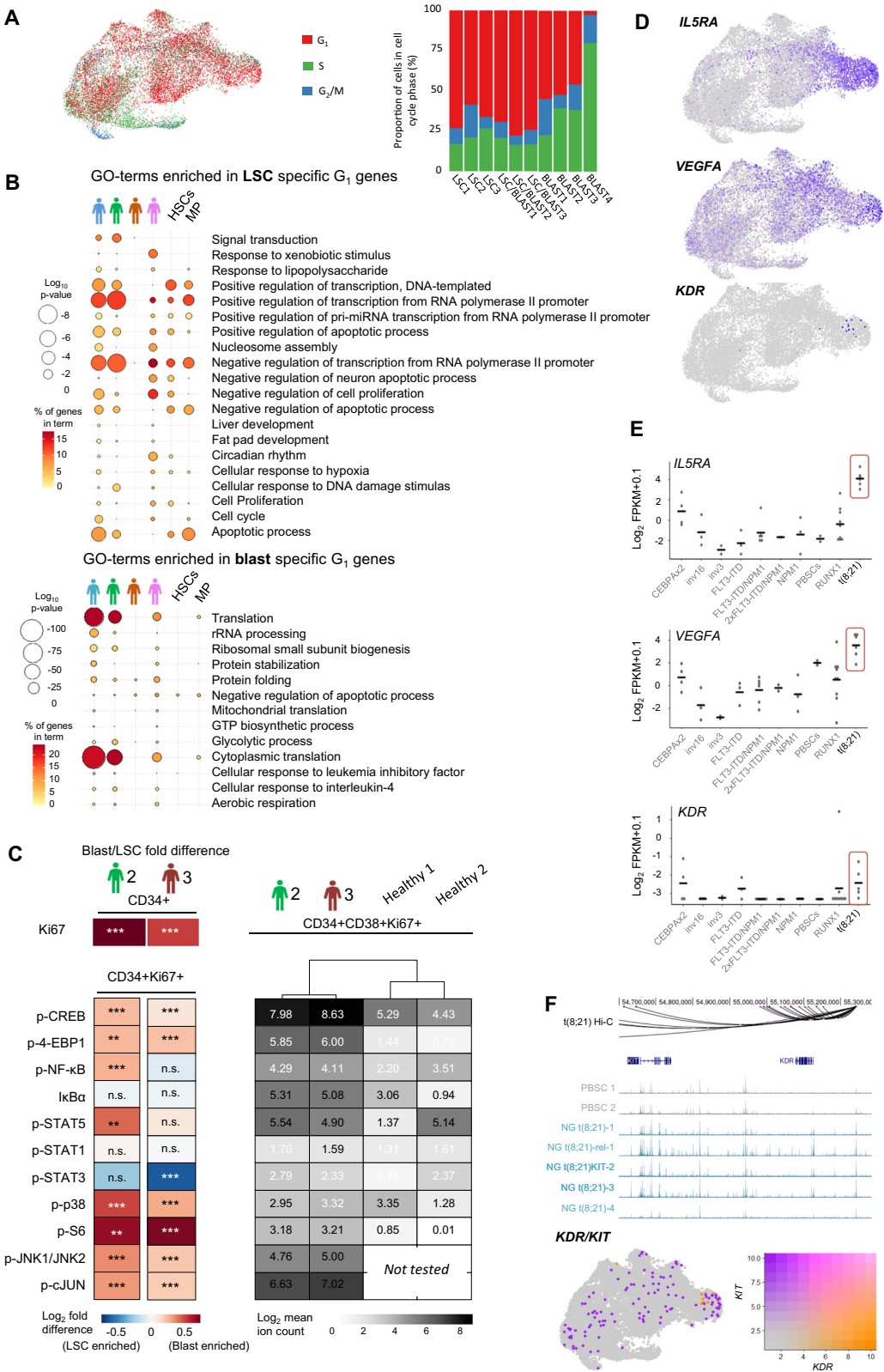

Our next experiment studied whether IL-5 and/or VEGF signalling were indeed capable of activating the growth of primary LSCs. To this end, we used different membrane tracking dyes to separately label purified LSCs and blasts from t(8;21) patient #2 peripheral blood grown together in cytokine-rich media with or without IL-5 and VEGF (Fig. 3H, I). Similar proportions of LSCs and blasts were detected at the end of each assay with or without IL-5/VEGF, comparable to the

proportion which were sorted and stained at the start, confirming the reliability of the membrane stains (Fig. 3I). The fidelity of the gates was also confirmed by staining known proportions of unsorted cells. Both LSCs and blasts proliferated during the experiment in response to the cytokines and small molecules present in the culture. However, after the addition of IL-5 and VEGF, proliferation as measured by EdU incorporation increased, from 73% to 80% in the blasts and from 73% to

**Fig. 2 | t(8;21) AML LSCs are differentially signalling responsive. A** The assigned cell cycle stage of each cell projected on to the UMAP plots and the proportion of cells in each cluster assigned to each cell cycle stage by their gene expression pattern. **B** Bubble plot showing enriched GO-terms generated from blast- or leukaemic stem cell (LSC)-specific genes from $G_0/G_1$ cells only from the individual t(8;21) patients, and from healthy haematopoietic stem cells (HSCs) and myeloid progenitors (MPs)[25], the colour scale indicates the % of genes in the GO-term which were found in the specific gene list, and the size of the bubble indicates the $\log_{10}$ $p$-value of the enrichment of the term as determined by GO-term enrichment analysis. **C** Heatmaps showing the $\log_2$ fold difference between mean ion counts in blasts and LSCs from two patients (left) and the $\log_2$ mean ion count in t(8;21) blasts and healthy bone marrow-derived haematopoietic stem cells (right) from mass cytometry/CyTOF. Ki67 is shown from total CD34+ cells, all other markers are shown from CD34 + Ki67+ cells. *P*-values for blast/LSC differences are indicated by n.s. >0.001, *<0.001, **<1e-5, ***<1e-10 using two-sided Student's *T*-tests. Patient 2 LSC $n = 414$, blasts $n = 4486$, patient 3 LSC $n = 6236$, blasts $n = 4229$. **D** Expression of *VEGFA, IL5RA* and *KDR* projected onto the UMAP plot, where blue indicates the normalised UMI count. **E** Normalised $\log_2$ (FPKM+0.1) of *IL5RA, VEGFA* and *KDR* in AML with different driver mutations and healthy CD34+ PBSCs[7]. Horizontal bars indicate the median of all samples. $N = 4$ CEBPAx2, 3 inv(16), 2 inv(3), 3 FLT3-ITD, 2 FLT3-ITD2x/NPM1, 3 NPM1, 2 PBSCs, 9 RUNX1 and 5 t(8;21) patients. **F** UCSC genome browser screenshots showing DNaseI in healthy CD34+ PBSCs (grey) and t(8;21) AML patients (blue)[7] at the *KIT* and *KDR* locus, with promoter capture Hi-C interactions shown at the top. Expression of *KIT* (purple) and *KDR* (orange) is projected onto the UMAP plot, where the colour indicates the relative expression of each gene alone or combined with a blend threshold of 0.2.

---

85% in LSCs. Results were the same regardless of which dye was used for which cell population (Supplementary Fig. 3H). Notably, the membrane dye was more variably detected with addition of IL-5 and VEGF - particularly for the LSCs - due to dilution following cell division.

To generate an unlimited source of human primary t(8;21) cells we developed a PDX generated from t(8;21) patient #5 who had relapsed with a KIT D816V mutation. This is - to our knowledge - the first PDX from a t(8;21) patient capable of serial re-engraftment[32]. Cells could be cultured ex vivo but did not form colonies. Upon secondary engraftment the PDX maintained the gene expression pattern of the original patient cells (Supplementary Fig. 4A) with a leukemia initiating cell frequency of >10$^{-3}$ (Supplementary Fig. 4B). As with the cell lines, addition of VEGF or IL-5 to cultured PDX cells stimulated growth, though with no additive effect when used together, whilst both inhibitors reduced growth (Fig. 4A–C). Healthy CD34+ cells showed no response to bevacizumab or benralizumab in the effective dose range observed for the t(8;21) cells (Fig. 4B, C, Supplementary Fig. 4C, D). We then tested inhibition of VEGFA and IL5RA in vivo by injecting PDX cells intra-femorally into NSG mice and treating the animals for 41 days with bevacizumab or benralizumab (Fig. 4D). The experiment was performed twice, analyzing 3–5 mice each time. Engraftment was measured by sampling peripheral blood after 92–99 days, and bone marrow was assayed at the endpoint from the injected (right) femur and the contra-lateral (left) leg to which AML cells have spread and grown. Fewer human CD45+ cells were found in peripheral blood samples from treated mice compared to vehicle only controls (Fig. 4E, F). All hCD45+ cells measured in peripheral blood were CD34+ and CD33+ showing that the cells underwent little or no myeloid differentiation. KDR and IL5RA positive cells were found predominantly in the LSC compartment of the recovered PDX cells (Fig. 4G, H), demonstrating that the PDX model faithfully recapitulates the phenotype of the primary cells from patients. KDR/IL5RA double positive LSCs were depleted or blocked from self-renewal by both inhibitors, with KDR single-positive LSCs further depleted by bevacizumab only (Fig. 4I, J and Supplementary Fig. 4E). We also noted a modest increase in the proportion of CD34-/CD11b+ cells indicative of more mature cells in the non-injected bone marrow with treatment (Supplementary Fig. 4F, G). Effects with benralizumab treatment showed the same trend as with bevacizumab but were smaller in magnitude. Whilst AML cells and surrounding tissues both produce VEGFA, IL-5 is only produced in small amounts in immunodeficient mice.

These results confirm that t(8;21) patient cells proliferate in response to VEGF and IL-5, preferentially in the LSC compartment. Taken together, these data show that LSC growth activation and self-renewal can be controlled by VEGF and IL-5 signalling.

**VEGF and IL-5 signals terminate at the AP-1 family of transcription factors**

We then asked how VEGF and IL-5 signalling exert their effects on LSC growth activation. Both signalling pathways are known to function via MAP-Kinase activation of the AP-1 family of transcription factors to control gene expression. We, and others, have shown that AP-1 is a critical regulator of growth and gene expression in t(8;21) AML as well as other subtypes[7,14,16] with both AP-1 and MAP-Kinase differentially active in proliferating blasts (Fig. 2C). To investigate this idea, we generated Kasumi-1[16] and SKNO-1 cell lines expressing a doxycycline-inducible, flag-tagged, broad range dominant negative FOS (dnFOS) peptide[33]. AP-1 binding to DNA is dependent on its assembly as a heterodimer of FOS and JUN family proteins but dnFOS blocks binding of all JUN family proteins to DNA via an acid domain. When induced, the peptide was largely localised to the cytoplasm, presumably sequestering JUN proteins (Supplementary Fig. 5A). Combining dnFOS induction with VEGF or IL-5 treatment drastically reduced growth rates compared to the control, negating stimulation of growth by either factor (Fig. 5A, B). Combination of bevacizumab and dnFOS did not show any additive effect (Fig. 5A, B). Treatment of Kasumi-1 cells with bevacizumab showed an almost complete ablation of FOS binding in chromatin as measured by chromatin immunoprecipitation followed by sequencing (ChIP-Seq) (Fig. 5C and Supplementary Fig. 5B). Induction of dnFOS only in both t(8;21) cell lines significantly reduced the growth rate as compared to an empty vector (EV) control (Fig. 5D, E and Supplementary Fig. 5C, D). Furthermore, dnFOS induction significantly reduced colony formation initially but increased comparative re-plating capacity (Fig. 5F, G) showing a shift in clonogenic frequency with proportionally more immature cells present after AP-1 was blocked. These results are in concordance with the reduced growth and increased self-renewal seen with bevacizumab and benralizumab treatment, and show that blocking AP-1 with dnFOS can be used to simultaneously inhibit both IL-5 and VEGF-stimulated growth. Our data demonstrates that VEGF and IL-5 signalling controls growth and self-renewal of LSCs upstream of AP-1.

**AP-1 orchestrates a shift in transcriptional regulation from an LSC to a blast pattern**

The data described above suggest that VEGF and IL-5 signalling activate AP-1 to kick-start LSC growth. Therefore, we next sought to understand how this circuit feeds into control of gene expression. To this end we performed DNaseI-seq, RNA-seq and ChIP-seq experiments for multiple transcription factors in Kasumi-1 cells with or without dnFOS induction to link AP-1 binding to the wider gene regulatory network (Fig. 6A, B). Experiments used a Kasumi-1 cell clone (Fig. 6A) expressing high levels of dnFOS in response to doxycycline (Supplementary Fig. 6A).

The comparison of LSC and blast open chromatin regions had shown that LSC-specific accessible chromatin sites were enriched in GATA motifs (Supplementary Fig. 1D). We noted that Kasumi-1 cells expressing dnFOS showed gain of chromatin accessibility associated with increased binding of GATA2 at distal chromatin sites (Fig. 6B and Supplementary Fig. 6B). Lost chromatin accessibility was associated with loss of binding of myeloid factors: FOS, RUNX1, RUNX1::ETO, C/EBPα and PU.1. To ask whether these factors are binding and lost in combination, we examined the ChIP signal across a union of gained

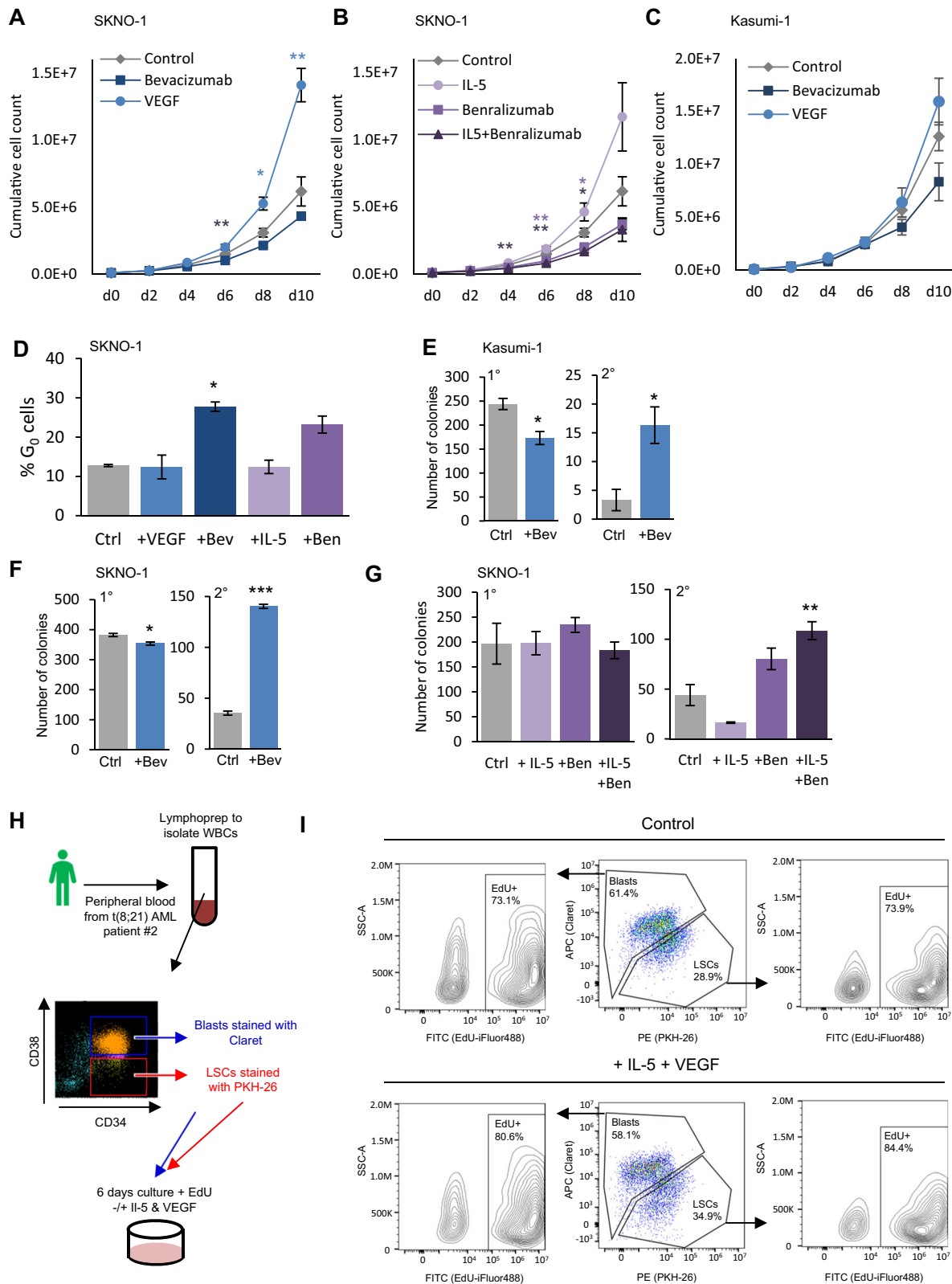

and lost binding sites for all six factors and performed a correlation analysis of the tag counts (Fig. 6C). This analysis examines whether the gained and lost peaks are the same for each factor, and indicated highly correlated binding patterns of RUNX1, RUNX1::ETO, FOS and C/EBPα at lost sites (Ctrl) and correlation of GATA2, C/EBPα and FOS binding at gained sites (Fig. 6C). Furthermore, an analysis of the motif spacing in the lost sites in each ChIP showed similar proximity of the

RUNX1, RUNX1::ETO and C/EBP consensus sequence to the AP-1 motif (Supplementary Fig. 6C). Motif enrichment analysis confirmed that the gained sites (dnFOS) were enriched for GATA, but not AP-1 motifs, whilst the lost sites (CTRL) shared AP-1, C/EBP and composite C/EBP:AP-1 motifs (Fig. 6D). C/EBP and AP-1 family members can heterodimerise[34] and *CEBPA* is repressed in t(8;21) AML[35] but the C/EBP family is required to maintain the viability of Kasumi-1 cells[10]. We

**Fig. 3 | Aberrant VEGF and IL-5 signalling in t(8;21) AML drives LSC growth.**
**A**–**C** SKNO-1 (**A**, **B**) and Kasumi-1 cells (**C**) were grown with bevacizumab, VEGF or media alone control (**A**, **C**) or IL-5, benralizumab, or IL-5 and benralizumab (**B**) for 10 days, with mean counts every two days indicated by the points, error bars indicate SEM. Controls are shared in A and B. $n = 3$ wells for + VEGF/+bevacizumab/+benralizumab/+IL-5+benralizumab, $n = 6$ for controls and +IL-5 (d6-10 = 5 for IL-5 due to sample loss) wells across 2 independent experiments. Statistical significance calculated by two-way ANOVA with Dunnett correction for multiple comparisons at each time point, p-values reported as compared to the control: SKNO-1 $p = 0.0048$ d6 + bevacizumab, $p = 0.0365$ d8 + VEGF, $p = 0.0087$ d10 + VEGF, $p = 0.0069$ d4 + IL-5 + benralizumab, $p = 0.0101$ d6 + benralizumab, $p = 0.0036$ d6 + IL-5 + benralizumab, $p = 0.0386$ d8 + IL-5 + benralizumab (**D**) Percentage of cells in $G_0$ as measured by Hoechst low and Pyronin Y negative following 24 h treatment with VEGF, bevacizumab (+Bev), IL-5 or benralizumab (+Ben) in SKNO-1, bars indicate the mean of 3 independent experiments and the error bars indicate SEM, $p = 0.0364$ for + bevacizumab by one way ANOVA with Bonferroni's multiple testing. **E**–**G** Primary (1°) and secondary (2°) replating colony forming assays +/− bevacizumab with Kasumi-1 (**E**) and SKNO-1 (**F**) and with IL-5, benralizumab or both in SKNO-1 (**G**), bars indicate the mean of 3 independent experiments and the error bars indicate SEM. $P = 0.0166$ for Kasumi-1 secondary colonies + bevacizumab, $p = 0.0166$ primary and $p < 0.0001$ secondary for SKNO-1 + bevacizumab (both two-sided unpaired Student's T-test) and $p = 0.0025$ for SKNO-1 + IL-5 + benralizumab (one way ANOVA with Bonferroni's multiple testing). **H** Schematic showing how the leukaemic stem cell (LSC) proliferation assay was conducted. **I** Flow cytometry plots identifying LSCs (stained with PKH-26, detected in the PE channel) and Blasts (stained with Claret, detected in the APC channel), with EdU (stained with iFluor488 and detected in the FITC channel) measured in each population separately. *$p < 0.05$, **$p < 0.01$, ***$p < 0.005$, for growth curves this is plotted in the same colour as the treatment group (**A**–**C**). Blue corresponds to VEGF/bevacizumab and orange to IL-5/benralizumab throughout, squares show bevacizumab, circles VEGF/IL-5, triangles IL-5 and benralizumab in (**A**–**C**). Source data are provided as a Source Data file.

therefore queried whether this result indicated that AP-1:C/EBP heterodimers were being disrupted, and whether this was a facet of the mechanism of action of AP-1 in t(8;21) AML. We therefore expressed a dnCEBP peptide in an inducible fashion as well[10,33]. Whilst dnCEBP expression led to loss of open chromatin containing AP-1 and RUNX1 binding motifs, it did not lead to gain of sites associated with GATA binding but instead gained AP-1 and RUNX1 binding sites (Supplementary Fig. 6D)[10]. Directly comparing the sites lost and gained with dnFOS and dnCEBP revealed similar but incompletely overlapping sites (Supplementary Fig. 6D, E) underpinning the discrepancy in motif patterns. Therefore, dnFOS and dnCEBP do not impact upon the same aspects of gene regulation, and whilst loss of AP-1 binding is associated with loss of C/EBPα binding, loss of AP-1 activity specifically contributes to a gain of GATA2 binding.

After induction of dnFOS expression, 226 genes were significantly up-regulated and 60 were down-regulated by at least 2-fold (Supplementary Fig. 6F, Supplementary Data 3). The comparison of binding alterations as measured by ChIP and gene expression changes showed that loss of FOS, RUNX1 and RUNX1::ETO binding led to both up and down-regulation of genes (Supplementary Fig. 6G). However, the acquisition of GATA2 binding was predominantly observed at elements associated with up-regulated genes. *GATA2* expression was also up-regulated by auto-regulation via increased binding to its enhancers (Supplementary Fig. 6H–I). GATA2 is a key regulator of stem cell maintenance[20] and its gene is expressed in LSCs[36] (Fig. 1). Based on these factors, although Kasumi-1 cells do not resemble quiescent LSCs, we measured whether the cell cycle block and the increase in GATA2 binding would lead to a reactivation of LSC-specific genes. To this end, we assessed transcription factor binding and histone modification at LSC or blast cell specific cis-regulatory elements and gene promoters as defined in Fig. 1 in the absence of AP-1 activity (Fig. 6E). This analysis showed a relative reduction in binding of FOS, RUNX1 and GATA2 at blast associated sites, whilst GATA2, RUNX1::ETO, PU.1 and FOS binding was increased at LSC-specific sites (Fig. 6E). We further found that in the Kasumi-1 cell line, LSC-specific gene promoters carried H3K27me3 but were also marked with H3K4me3 indicating that they are in a bivalent or poised chromatin conformation[37] (Fig. 6F, G). Together these data show that inhibiting AP-1 reactivates poised LSC genes and silences blast genes through a shift in FOS and PU.1 to GATA2 sites and loss of RUNX1 particularly from AP-1 and C/EBP sites.

## AP-1 is required for maintenance of the blast program
To confirm the notion that inhibition of AP-1 reactivates an LSC gene expression signature, in primary t(8;21) AML cells containing authentic LSCs, we transduced dnFOS or an EV control into PDX cells and into healthy CD34+ PBSCs. We sorted the dnFOS/GFP expressing cells following transduction and dox induction (Fig. 7A and Supplementary Fig. 7A). In PDX cells 160 genes were up-regulated and 129

genes down-regulated by at least 2-fold (Fig. 7B). In healthy cells fewer genes were de-regulated and to a lesser extent (Fig. 7C, Supplementary Fig. 7B, Supplementary Data 4). Moreover, healthy cells did not show a phenotypic response to dnFOS in colony forming assays (Supplementary Fig. 7C). To ask how dnFOS induction impacted LSC and blast gene expression programs, we performed Gene Set Enrichment Analysis (GSEA) based on the t(8;21) LSC and blast gene signature defined in Fig. 1. The genes up-regulated in the dnFOS-expressing PDX cells were enriched for LSC genes, and the down-regulated genes for the blast signature, which was not the case for healthy PBSCs (Fig. 7D, E).

## The signalling response of t(8;21) cells operates within a RUNX1::ETO dependent regulatory circuit
RUNX1::ETO is required for the maintenance of the leukemic state in t(8;21) cells as its depletion activates a C/EBPα-dependent myeloid differentiation program[12,13,38,39]. The above data show that AP-1 is also required to support growth of t(8;21) cells but AP-1 family gene expression is a feature of most subtypes of AML (Supplementary Fig. 8A). Expression of RUNX1::ETO notably leads to the activation of *JUN*[40–42]. Of the six AP-1 family genes most highly expressed in t(8;21) AML we found that all except *FOSB* showed significantly higher expression in LSCs and/or the LSC transition population (Supplementary Fig. 8B). We therefore asked how the VEGF and IL-5 signalling pathways, together with AP-1 family members, are regulated with respect to the driver oncoprotein RUNX1::ETO. We investigated gene expression and AP-1 binding with and without RUNX1::ETO depletion in a Kasumi-1 cell line carrying an inducible shRNA targeting RUNX1::ETO. These knockdown experiments showed that *JUNB*, *JUN* and *JUND* were up-regulated in the presence of RUNX1::ETO, whilst expression of *FOS* and *FOSB* were not (Fig. 8A). *VEGFA* was also down-regulated with RUNX1::ETO knockdown (Fig. 8A), similarly, in SKNO-1 targeted with RUNX1::ETO siRNA both *VEGFA* and *KDR* are down-regulated[35]. Following RUNX1::ETO knockdown, cells were no longer able to grow in response to VEGF stimulation, nor could VEGF stimulation rescue the impact of knockdown on growth (Fig. 8B). FOS shows a large overlap in binding sites with JUN and JUND[38] in wild-type Kasumi-1 cells as shown by ChIP-Seq (Supplementary Fig. 8C). However, JUN and FOS proteins, like the mRNAs, behaved differently with respect to RUNX1::ETO depletion, as exemplified by FOS and JUND. JUND binding was decreased across all binding sites after knockdown of RUNX1::ETO (Fig. 8C). In contrast, FOS was lost from distal cis-regulatory elements containing AP-1 motifs and re-distributed to promoters with accessible chromatin and bound POLII (Fig. 8C and Supplementary Fig. 8D, E). Most FOS and RUNX1::ETO binding sites, whilst responsive to oncoprotein depletion, do not overlap. However, many FOS sites overlapped with RUNX1 bound sites (Supplementary Fig. 8F). Together these data show that both AP-1 expression and localisation are orchestrated by RUNX1::ETO and further regulated by VEGF and IL-

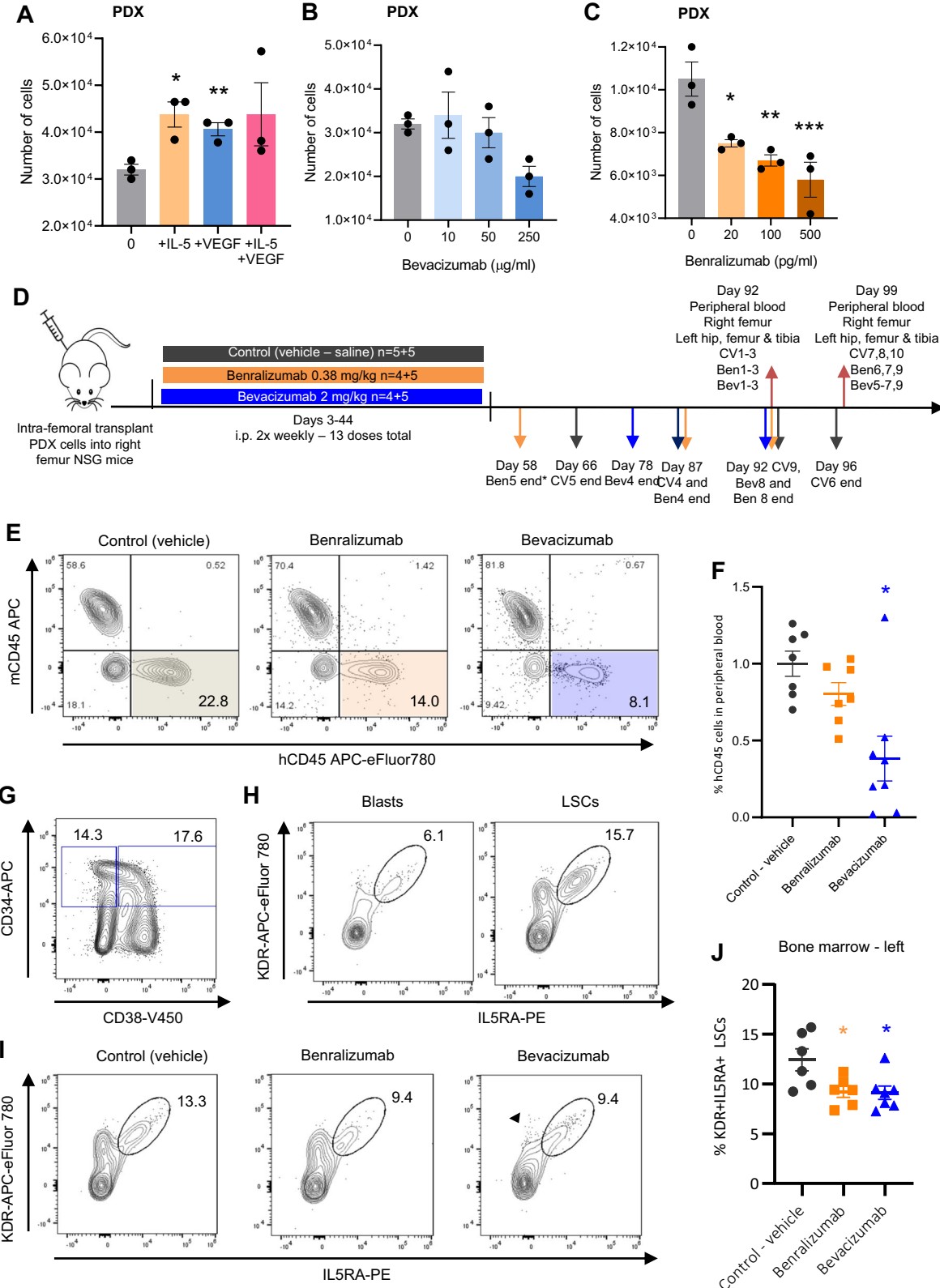

5 signalling. We further found that the signalling responsive histone modification H3K9acS10P[43] was globally reduced following RUN-X1::ETO knockdown despite an increase in total H3K9ac (Supplementary Fig. 8G)[12]. Although not directly associated with the altered FOS binding, the loss of this histone modification implies signalling to chromatin indeed relies upon RUNX1::ETO.

AP-1 gene expression and its binding to DNA are normally only detected at a substantial level in the presence of active signalling[44]. To

investigate whether the expression of *VEGFA* and *IL5RA* are signalling or RUNX1::ETO responsive and thus form a regulatory circuitry, we examined their cis-regulatory regions. We assembled DNaseI-seq data from t(8;21) AML patients together with the above-described DNaseI-seq and ChIP-seq data for myeloid transcription factors from Kasumi-1 cells with dnFOS. Two results were noteworthy: (i) All three genes showed a DHS at their promoters in healthy PSBCs (Fig. 8D–F) and in purified HSCs[19] suggesting that their promoters were poised for

**Fig. 4 | VEGFA and IL5RA inhibitors reduce PDX proliferation. A−C** t(8;21) PDX cells were grown in vitro for 6 days with or without IL-5 and/or VEGF(165) (**A**), with 3 doses of bevacizumab (**B**) or with 3 doses of benralizumab (+10 ng/ml IL-5) (**C**) and the resulting cells counted. Control/0 bevacizumab sample in (**A**) and (**B**) is the same as experiments were performed in parallel. Bar height shows the mean of 3 wells and error bars indicate SEM, $p = 0.0159 +$ IL-5, $p = 0.0090 +$ VEGF in (**A**) and $p = 0.0214$ 20 pg/ml, $p = 0.0057$ 100 pg/ml, $p = 0.0015$ 500 pg/ml benralizumab in (**B**, **C**). **D** Schematic showing how PDX dosing and sampling were conducted in vivo, analyses were performed on mice reaching day 90+. All mice taken prior to the fixed end point had weight loss or leg tumours associated with the PDX except * which had an enlarged thymus. **E** Representative contour plots showing the human and mouse CD45 positive cells by flow cytometry in peripheral blood at day 92 post-injection. **F** Percentage of human CD45 positive cells in peripheral blood at end point indicated in (**D**), normalised to the mean of the vehicle control mice for experiment 1 (CV1-3, + benralizumab (Ben) 1-3, + bevacizumab (Bev) 1-3) and experiment 2 (CV7-10, Ben6-9, Bev5-9), $n = 7$ mice for control, 7 mice benralizumab,

8 mice bevazicumab. $P = 0.0997$ benralizumab, $p = 0.0035$ bevacizumab. **G**, **H** Representative contour plots showing the relative populations of hCD45 + CD34 + CD38 + /− cells (**G**) and KDR and IL5RA positivity of hCD45+/CD34+/CD38+ blast cells and hCD45+/CD34+/CD38- LSCs (**H**) in control left bone marrow at day 92 post-injection. **I** Representative contour plots showing the KDR and IL5RA positivity of hCD45+/CD34+/CD38− LSCs in treated or control left bone-marrow at day 92 post-injection. **J** Percentage of KDR and IL5RA positive hCD45+/CD34+/CD38− LSCs in left bone marrow at day 92/99 post-injection, $p = 0.0359$ benralizumab, $p = 0.0320$ bevacizumab, $n = 6$ mice for control, 6 mice benralizumab, 7 mice bevacizumab. **F, J** Horizontal and error bars show mean and SEM respectively of the mice in each treatment group, *$p < 0.05$, **$p < 0.01$, ***$p < 0.005$ using two-tailed unpaired Student's $t$-tests vs control (**A**), one-way ANOVA with Bonferroni's multiple comparisons (**B**, **C**) and two-tailed unpaired Welch's $t$-test vs vehicle controls (**F**, **J**). Blue corresponds to bevacizumab and orange to benralizumab throughout. Source data are provided as a Source Data file.

expression; (ii) the *VEGFA* and *KDR* promoters were bound by FOS whose binding was responsive to dnFOS and RUNX1::ETO depletion, linking gene expression control directly to factor binding.

At the *ILSRA* locus, two specific DNaseI peaks were detected in t(8;21) AML patients (Fig. 8D) but not in healthy PBSCs. No ChIP or DNaseI-signal was detected here in Kasumi-1 cells. A motif search in the DNaseI hypersensitive sites (DHSs) from primary cells revealed GATA, AP-1 and RUNX binding motifs (Fig. 8D and Supplementary Fig. 8H). Regulation of *VEGFA* and *KDR* was more complex, with multiple peaks and broad regulatory regions in both genes (Fig. 8E, F). None of the DHSs were exclusive to LSCs, suggesting that signalling-responsive transcription factor binding activity controls specificity of expression. After inhibition of AP-1 we observed loss of chromatin accessibility and FOS binding at these DHSs, and at some peaks loss of RUNX1 binding as well. After shRUNX1::ETO induction, both FOS binding and the H3K9acS10P were largely unchanged at these sites, despite *VEGFA* expression decreasing after RUNX1::ETO knockdown (Fig. 8A). Taken together, our data show that *VEGFA*, *KDR* and *IL5RA* are regulated by a complex interplay of activating and repressing transcription factors operating within the context of a primed and signalling responsive chromatin landscape.

## Discussion

In this study, we have shown that t(8;21) LSCs specifically utilise VEGF and IL-5 signalling to promote growth. VEGF and IL-5RA are aberrantly expressed in t(8;21) LSCs as part of a regulatory circuit involving the driver oncogene RUNX1::ETO and the AP-1 complex as a mediator of signalling. This interplay forms a balanced feed-forward loop with RUNX1::ETO at the apex (Fig. 8G). RUNX1::ETO blocks differentiation by down-regulating *CEBPA*[35] and disrupting PU.1 and RUNX1 driven control of myelopoiesis[12,45]. Simultaneously, RUNX1::ETO, when expressed on its own, blocks the cell cycle[42] which is overcome by active signalling causing the up-regulation and post-translational activation of the AP-1 complex[16]. Furthermore, AP-1 is required for myeloid differentiation as its inhibition up-regulates *GATA2* expression and shifts cells to a more immature state. From our previous work[16] we know that inhibiting AP-1 activity causes a cell cycle block in t(8;21) cells, as cell cycle genes which are also bound by RUNX1-ETO and RUNX1 are down-regulated. AP-1 activation orchestrates changes in the transcriptional program through altering C/EBPα, PU.1, RUNX1 and GATA2 binding patterns, leading to a reversible silencing of LSC genes and the activation of blast genes, preserving self-renewal capacity whilst allowing cell expansion.

*VEGFA*, *KDR* and *IL5RA* are not normally expressed in myeloid or stem cells but show a primed chromatin structure in healthy HSCs, with the promoters being hypersensitive and ready to be expressed[19]. Each of these genes is a target for AP-1 mediated signalling transduction in established AML cells, but AP-1 is also involved in co-opting

*VEGFA* into supporting the growth of non-myeloid leukemic cells[46]. *VEGFA*, *KDR* and *IL5RA* are GATA2 targets, whereby GATA2 further co-operates with AP-1[47] and is specifically expressed in LSCs which are poised to cycle[48]. An important result from our study is therefore that the exact signalling pathways employed by LSCs are highly subtype specific, relying on the specific interplay of the driver mutation with the stem cell program. Using published scRNA-seq data we can confirm our result, with a cluster of cells co-expressing *IL5RA*, *VEGFA* and *GATA2* detected in the t(8;21) AML sample[49]. During embryonic development and thereafter, *VEGFA* and *KDR*, which are part of the endothelial gene expression program, are repressed by RUNX1[50] RUNX1::ETO disrupts the action of wild-type RUNX1 on *VEGFA/KDR*[51] and endothelial gene expression remains elevated[52,53]. In CEBPA double mutant AML (CEBPAx2), RUNX1 expression is down-regulated[10] as well and as a result *VEGFA*, *KDR* and *IL5RA* are still expressed but at a lower level than in t(8;21). Moreover, inspection of LSC and blast cell single cell data[10] demonstrated that this type of AML also activates a specific, but different cytokine receptor, *CSF2RB*, the common subunit for the IL-5, IL-3 and GM-CSF receptors (Supplementary Fig. 8I, J) suggesting that ectopic pathway expression is used by LSCs in more than one AML sub-type.

Activation of ectopic signalling pathways in LSCs leads to the re-generation of full-scale leukemia with the signals coming from the environment in which they reside. IL-5 is normally produced by eosinophils, mast cells and stromal cells, whilst VEGF-signalling originates from the vascular niche and in t(8;21) from the AML cells themselves. VEGF also contributes to engineering of the niche by leukemic cells to better support their growth[54–56]. In this scenario, relapse is inevitable as LSCs are ready to respond to signals, which will eventually arrive. It has been shown that LSCs undergo a transient amplification after chemotherapy[57]. Therapy therefore needs to target rapidly growing blast cells and block signalling to prevent re-entry of LSCs into the cell cycle. In t(8;21) AML this may be achieved by repurposing the FDA-approved monoclonal antibodies bevacizumab and/or benralizumab. Inhibition of VEGF by bevacizumab has been previously trialled in AML to block remodelling of the niche but only 2 core-binding factor AML patients of unknown genotype were included[58] and the results overall were therefore inconclusive. In summary, our work highlights the importance of studying the fine details of AML sub-type specific gene regulatory networks impacting on specific mechanisms of growth control to find the right therapeutic targets to prevent relapse.

## Methods
### Experimental models

**Primary cultures.** Human tissue was obtained with the required ethical approval from the NHS National Research Ethics Committee and informed consent from patients. Patient bone marrow biopsies were obtained, and the AML cells purified using lymphoprep

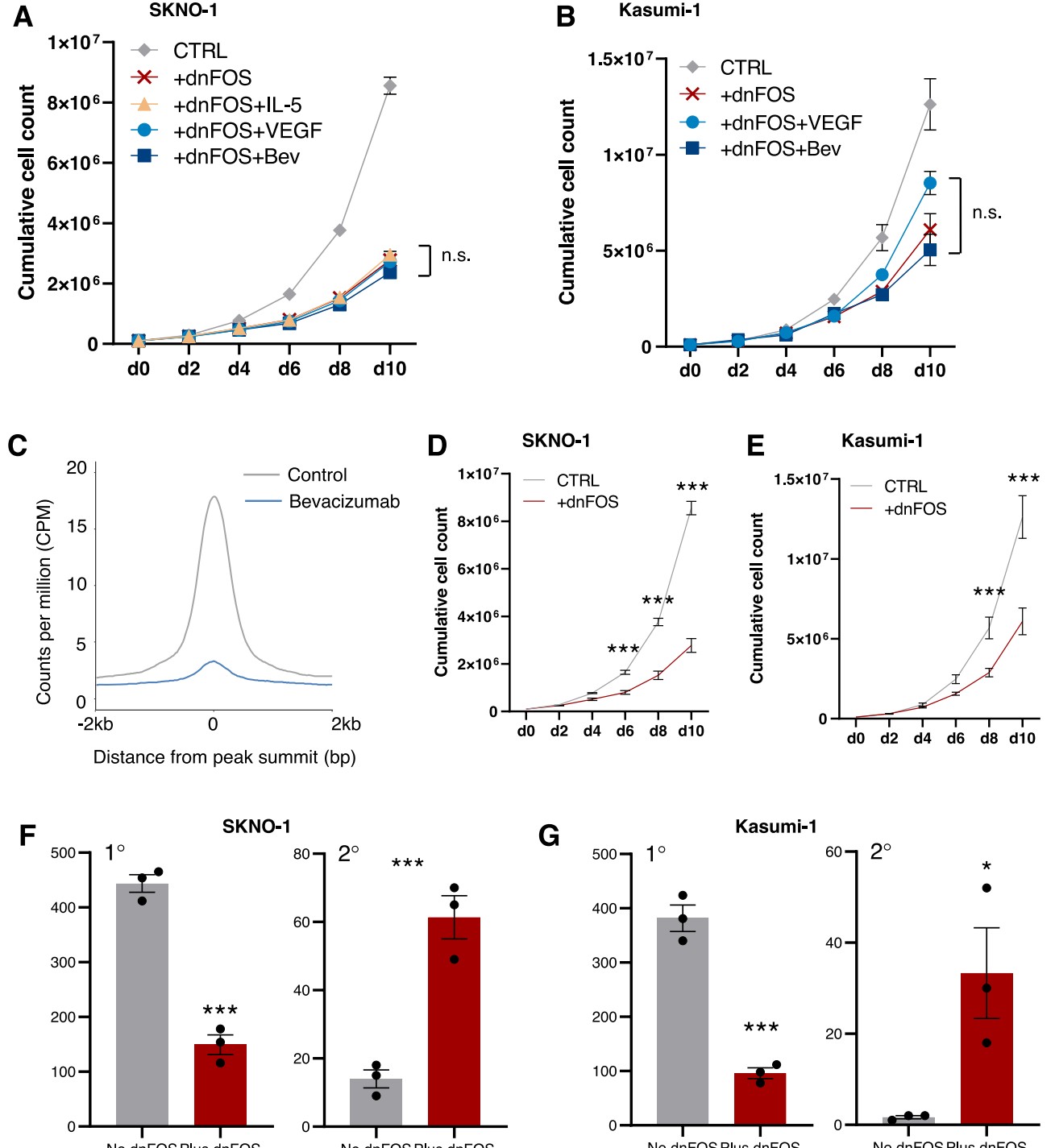

**Fig. 5 | VEGF and IL-5 signals terminate at the AP-1 family of transcription factors. A, B** Growth curves were performed by growing SKNO-1 (**A**) or Kasumi-1 (**B**) cells for 10 days, counting and passaging every 2 days. Cells were grown with induction of dnFOS by doxycycline in conjunction with IL-5 (SKNO-1 only), VEGF-165 or bevacizumab. Each point represents the mean of three experiments except Kasumi-1 control and +dnFOS $n = 6$ wells across 2 independent experiments, and error bars show SEM. The control curves are the same as in Fig. 3 as experiments were performed in parallel and shown again for clarity. No significant differences were found at any time point comparing +dnFOS with any treatment group, with two-way ANOVA with Dunnett correction for multiple comparisons at each time point. Grey diamonds show control, red X +dnFOS, orange triangles +IL-5, light blue circles +VEGF, dark blue squares +bevacizumab (+Bev) (**C**) Histogram showing the average normalised FOS ChIP signal across the union of all peaks, +/−2 kb of the summit in Kasumi-1 cells with and without bevacizumab. **D, E** Growth curves were

performed by growing SKNO-1 (**D**) or Kasumi-1 (**E**) cells for 10 days, counting and passaging every 2 days. Cells were grown with or without dnFOS induced by doxycycline. Data as in (**A**, **B**). $P = 0.0010$ d6, $p < 0.0001$ d8 and $p < 0.0001$ d10 for SKNO-1 + dnFOS; $p = 0.0009$ d8 and $p < 0.0001$ d10 for Kasumi-1 +dnFOS. **F, G** Colony forming unit assays were performed by plating SKNO-1 (**F**) or Kasumi-1 (**G**) cells in methylcellulose with or without doxycycline to induce dnFOS. The number of colonies were counted after 10 days (left) and cells were replated to form secondary colonies which were again counted after 10 days (right). Bars indicate the mean of three independent experiments, error bars show SEM. $P = 0.0003$ primary and $p = 0.0023$ secondary colonies in SKNO-1, $p = 0.00004$ primary and $p = 0.0336$ secondary colonies in Kasumi-1. $*p < 0.5$ and $***p < 0.005$ using two-tailed unpaired Student's $T$-tests vs controls. Grey corresponds to control and red to +dnFOS in (**D–G**). Source data are provided as a Source Data file.

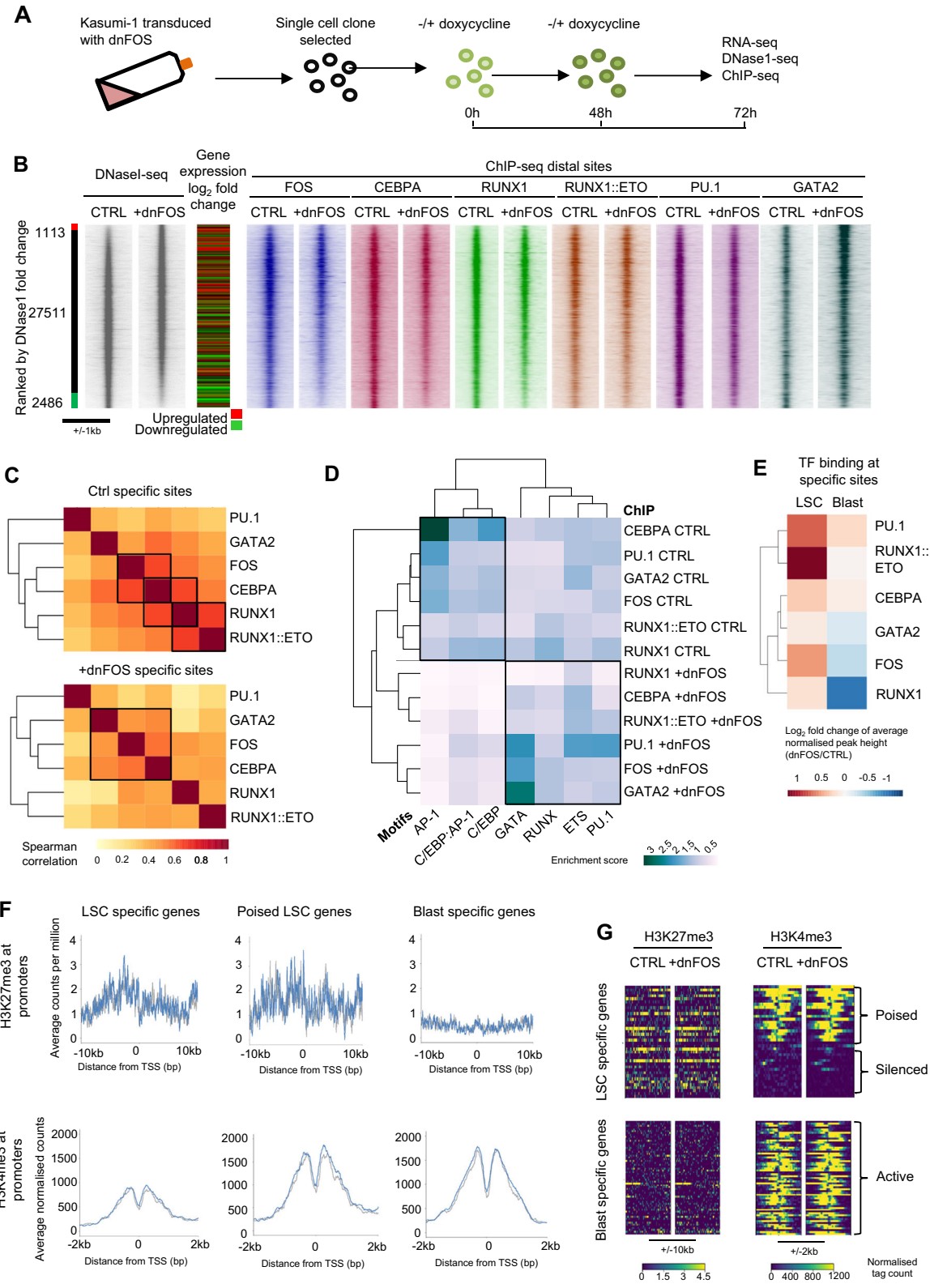

followed by CD34 MACS bead enrichment. Patient mutation details are in Table 1. Primary cells and PDX cells (patient 5 only) were cultured on human mesenchymal stem cells, in SFEMII (StemCell Technologies) supplemented with 1% Pencillin/Streptomycin, 1 µM UM729 (Stemcell Technologies), 750 nM SR1 (Stemcell Technologies), 150 ng/ml SCF, 100 ng/ml TPO, 10 ng/ml FLT3, 10 ng/ml IL3, 10 ng/ml GM-CSF (all cytokines from Peprotech). Where primary

cells were frozen prior to use, they were allowed to recover for a week before performing phenotypic assays but sorted directly from defrost for gene expression analysis. Healthy CD34+ cells (Amsbio) were cultured in SFEMII with StemSpan CD34 Expansion Supplement (Stem Cell Technologies) and 500 nM UM729 for 1 week, then moved into the t(8;21) media for 24 h prior to setting up assays.

**Fig. 6 | AP-1 orchestrates a shift in transcriptional regulation from an LSC to a blast pattern. A** Schematic showing how dnFOS was induced in Kasumi-1 cells **B** DNase1 was performed with and without dnFOS induced by doxycycline in the Kasumi-1 cell line, ranked by fold change of the tag count at distal peaks and represented as density plots (+/1 kb of the summit). The red bar indicates dnFOS specific sites and the green bar control specific sites where the normalised tag-count of specific sites is at least two-fold different. ChIP data from FOS, CEBPA, RUNX1, RUNX1::ETO, PU.1 and GATA2 with and without dnFOS were plotted on the same axis across the same window **C** Specific sites were calculated for the ChIPs shown in (**A**) where the normalised tag-count is at least two-fold different in a pairwise comparison of dnFOS against control. The normalised tag count was measured in a peak union generated from control or dnFOS specific sites from all ChIPs and the Spearman correlation calculated which is plotted as a heatmap with

hierarchical clustering. **D** A motif enrichment score was calculated based on motif frequency in the specific gained (dnFOS) and lost (CTRL) sites calculated in (**B**) and plotted as a heatmap with hierarchical clustering. **E** Heatmap with hierarchical clustering showing the $\log_2$ fold change between the normalised average peak height of ChIP-seq in Kasumi-1 with dnFOS vs controls at LSC-specific and blast-specific ATAC sites. **F** Average profiles were generated from the CPM normalised tag counts of ChIP for H3K27me3 (+/− 10 kb from the transcription start site (TSS)) and H3K4me3 (+/− 2 kb from the TSS), at the promoters of t(8;21) LSC or blast specific genes with or without induction of dnFOS in Kasumi-1 cells, poised LSC genes are those with both H3K27me3 and H3K4me3 at their promoter. **G** Density plots showing the signal at each of the sites used in (**F**), with active, silenced and poised genes indicated.

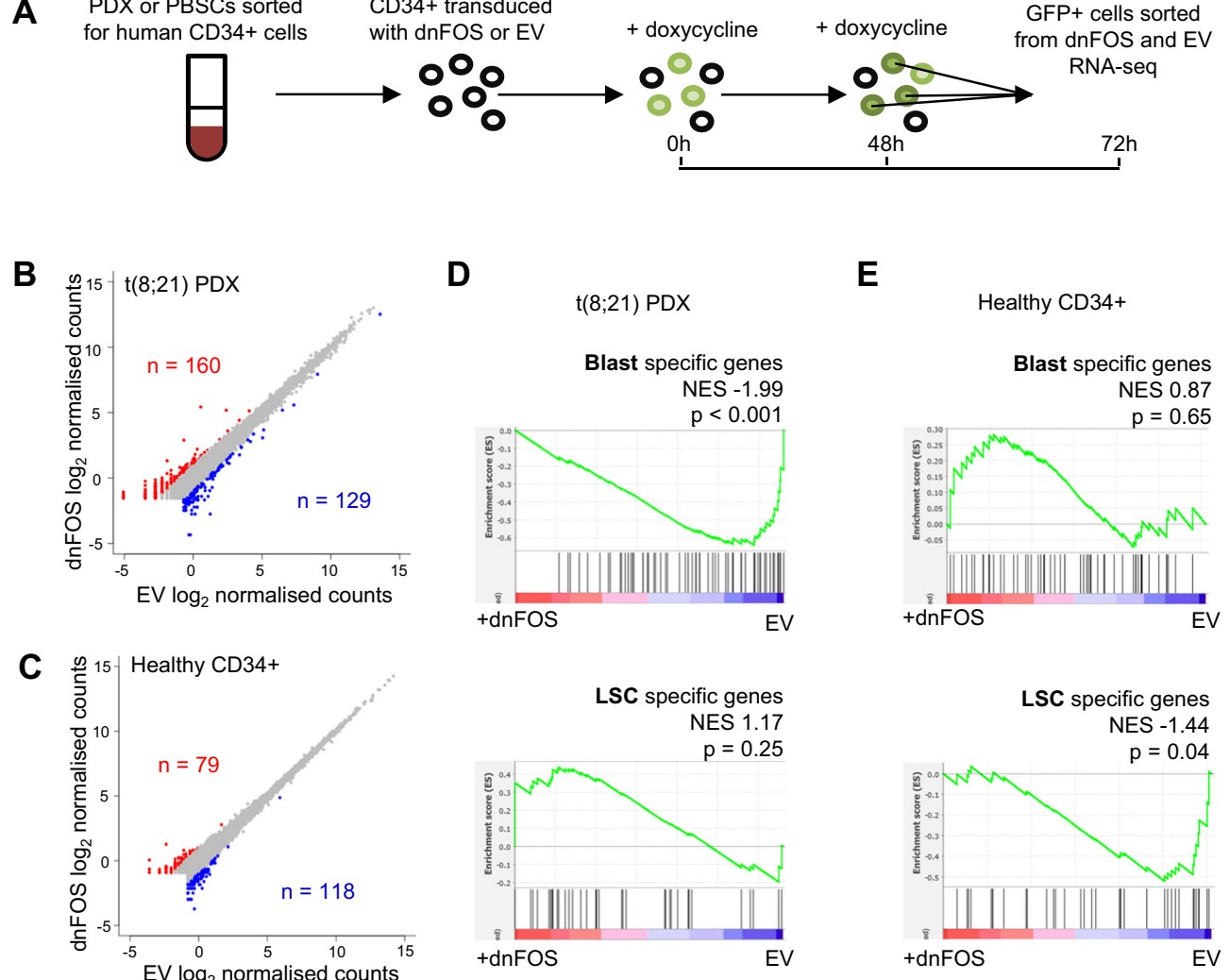

**Fig. 7 | AP-1 is required for maintenance of the blast program. A** Schematic showing how dnFOS was induced in primary cells **B**, **C** RNA-seq was performed in PDX cells and healthy CD34+ PBSCs following induction of dnFOS or the EV control, gene expression is shown as a scatter plot of the $\log_2$ counts, with the genes up-regulated by dnFOS highlighted red and the down-regulated genes highlighted in

blue. **D**, **E** GSEA was used to compare blast and LSC specific genes identified from scRNA-seq with the ranked fold change gene expression from the PDX (**D**) and healthy CD34+ cells (**E**), comparing dnFOS to EV. NES shows the normalised enrichment score from the GSEA and the nominal *p*-value as calculated by GSEA software.

**Cell lines.** Kasumi-1 (RRID: CVCL_0589; male; ACC 220), SKNO-1 (RRID: CVCL_2196; male; ACC 690), MOLM14 (ACC 777), MV4-11 (ACC 102), U937 (ACC 5) and HEK293T (ACC 305) cells were all obtained from DSMZ and were routinely maintained in RPMI1640 medium or DMEM (HEK293T) supplemented with 10% or 20% FBS (SKNO-1), 2 mM L-Glutamine and 1% Penicillin/Streptomycin. SKNO-1 cells were

additionally supplemented with 10 ng/ml GM-CSF. All cells were incubated at 37 °C in a humidified 5% $CO_2$ incubator.

## Method details
**Plasmid generation.** Generation of the dnFOS plasmid was previously described[16] - dnFOS was amplified from cDNA provided by Charles

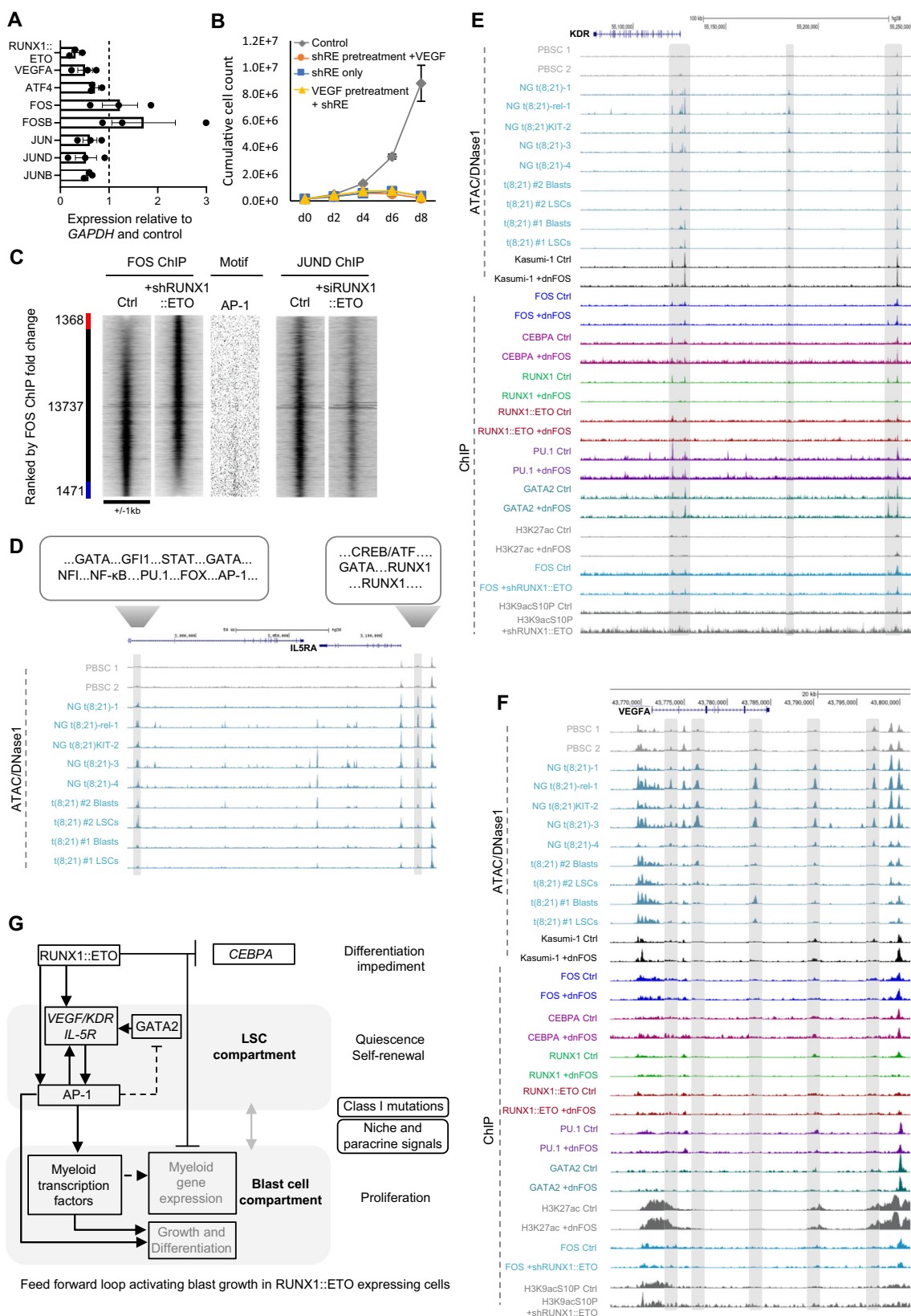

Figure G caption: Feed forward loop activating blast growth in RUNX1::ETO expressing cells

Vinson[33] with SalI and NotI restriction site overhangs. Using these restriction sites, the fragment was ligated into pENTR2B (Addgene) and then recombined into pCW57.1 (Addgene). The empty vector was pCW57.1 alone. The shRUNX1::ETO plasmid was generated with XhoI and EcoRI restriction site overhangs. Using these restriction sites, the

fragment was ligated into tRMPVIR (Addgene) plasmid. The shRNA sequence is 5′-AAACCTCGAAATCGTACTGAGA-3′. Plasmids were selected and propagated in DH5α competent cells prior to maxiprep using NucleoBond Xtra Midi EF kit and then lentiviral production. All unique biological materials are available from the authors upon request.

**Fig. 8 | The signalling response of t(8;21) cells operates within a RUNX1::ETO dependent regulatory circuit. A** qRT-PCR showing the change in expression of *RUNX1::ETO, VEGFA* and the most highly expressed AP-1 members after shRUNX1::ETO knockdown, relative to *GAPDH* and the no knockdown control. Bars indicate the average of 3 independent knockdown experiments, error bars show SEM, the vertical dashed line indicates no change in expression. Source data are provided as a Source Data file. **B** Growth curves were performed by growing Kasumi-1 cells for 8 days, counting and passaging every 2 days, following 2 days of pretreatment with either doxycycline to induce shRUNX1::ETO (orange and blue) or with VEGF-165 (yellow). Cells were grown with induction of shRUNX1::ETO by doxycycline in conjunction with VEGF-165 from d0. Each point represents the mean of three wells, and error bars show SEM. No significant differences were found at any time point comparing shRUNX1::ETO expressing cells (orange, blue, yellow). **C** ChIP for FOS was performed with and without shRUNX1::ETO induced by doxycycline in the Kasumi-1 cell line, ranked by fold change of the tag count at all peaks and represented as density plots (+/1 kb of the summit). The red bar indicates shRUNX1::ETO specific sites and the blue bar control specific sites where the normalised tag-count of specific sites is at least two-fold different. ChIP for JUND with siMM (Ctrl) or siRUNX1::ETO[38] and AP-1 motif frequency is plotted along the same axis across the same window. **D**–**F** UCSC genome browser screenshots showing ATAC/DNaseI in healthy CD34+ PBSCs and t(8;21) AML patients[7] at the *IL5RA* locus, with the transcription factor binding motifs in the t(8;21) specific peaks indicated (**D**), and additionally showing DNaseI and ChIP in Kasumi-1 +/− dnFOS, and +/− shRUNX1::ETO at the *KDR* (**E**) and *VEGFA* (**F**) loci with the t(8;21) specific peaks indicated by grey bars. **G** Model showing how AP-1 activated by signalling activates blast cell growth in t(8;21) AML.

**Lentivirus production and cell transduction.** Lentivirus was produced in HEK293T cells using calcium phosphate co-precipitation of the target plasmid and packaging vectors TAT, REV, GAG/POL and VSV-G at a mass ratio of 24 µg: 1.2 µg: 1.2 µg: 1.2 µg: 2.4 µg per 150 mm diameter plate of cells. Viral supernatant was harvested after 24, 36, 48 and 60 h then concentrated by ultracentrifugation at 25,000 × G for 1 h 45 min at 4 °C. Concentrated virus was then transduced into cell lines or primary cells with 8 µg/ml polybrene via spinoculation at 1500 × G for 45 min. Media was refreshed after 12 h. To generate clones, cell lines underwent puromycin selection (1 µg/ml) for 5 days and were then sorted for single cells by FACS.

**Growth curves.** For growth curves, cell lines were counted using trypan blue and passaged every 2 days, seeding cells at the original concentration. Cells were grown with 10 ng/ml IL-5 (Peprotech), 50 ng/ml VEGF-165 (Peprotech), 10 µg/ml Bevacizumab (Selleckchem) and/or 500 pg/ml Benralizumab (AstraZeneca). Where appropriate, doxycycline induction of transduced cells was at 2 µg/ml. Growth curves were not performed in the PDX, instead the cells were just counted at day 6 after seeding.

**Colony forming assays.** For colony assays, cells were grown for 24 h with the treatment to be tested, then seeded into H4100 MethoCult (Stem Cell) with RPMI1640 and 10% FBS, and the treatment to be tested including doxycycline as appropriate. Patient-derived cells were seeded into MethoCult Express (Stem Cell) Kasumi-1 were seeded at 2000 cells per dish, SKNO-1 were seeded at 5000 cells per dish and patient cells were seeded at 1000 cells per dish. Colony assays were counted after 10 days, except for patient-derived colonies which were assessed after 20 days.

**Flow cytometry/FACS.** Flow cytometry was carried out on a Cyan ADP (Beckman Coulter) using antibodies against CD309-APC (KDR, Cat# 130-117-984 Miltenyi Biotec) and CD125-biotin (IL5RA, Cat# 130-110-543 Miltenyi Biotec) followed by streptavidin-PE-Cy7 (Cat# 25-4317-82 ThermoFisher) for cell lines, and on an Attune NxT (Thermo Fisher) using antibodies against 1: hCD45-FITC, CD34-APC Cat# 130-120-519, CD38-V450 Cat# 646851 BD Biosciences, VEGFR-APC-vio770 Cat# 130-117-987 and IL5RA-PE Cat# 130-110-602, 2: hCD45 APC-eFluor780 Cat# 47-0459-42 ThermoFisher, CD34-PE Cat# 130-120-515 and mCD45-APC, or 3: CD33-BV421, CD11b-APC Cat# 130-091-241, CD34-PE and hCD45-APC-eFluor780 for PDX cells. All antibodies from Miltenyi Biotec unless otherwise stated. Cells were resuspended in 100 µl MACS buffer (PBS + 2 mM EDTA + 0.5% BSA) and all antibodies were added at 1:100, with staining for 30 min at 4 °C. Compensation was set up using cells and/or compensation beads. For $G_0$ assays, cells were incubated in 10 µg/ml Hoechst33342 for 45 min at 37 °C and then 5 µl of 100 µg/ml Pyronin Y was added for a further 15 min following which cells were kept on ice until analysis on the flow cytometer. Analysis was carried out on FlowJo v10.

FACS was carried out using a FACS Aria (BD) with antibodies from BD Biosciences. LSCs and blasts were identified and sorted using 7-AAD Cat# 559925 and lineage cocktail-FITC Cat# 340546 to select lineage-negative viable cells, followed by CD34-PE-Cy7 Cat# 348811 positive cells and gating CD38-V450 Cat# 646851 positive blasts and negative LSCs, see gating strategy in Supplmentary Fig. 1A. dnFOS transduced/induced PDX were gated for viability on forward/side scatter and sorted for GFP+ as compared to a non-transduced population. dnFOS transduced cell lines were sorted based on forward/side-scatter only to single cells.

**CyTOF panel design and in-house labelling of purified antibodies.** The AML CyTOF panel was designed to include cell markers specific for myeloid blasts and cell signalling markers of interest. For most of the targets, antibodies were acquired in pre-conjugated format from the Standard BioTools catalogue. For other targets (CD117, Cat# 313202 BioLegend, pJNK1/JNK2 Cat# 700031 ThermoFisher, p-cJUN Cat# PA5-104747 ThermoFisher, beta2-microglobulin Cat# 316302 Biolgend, CD298 Cat# 341712 Biolegend) we performed in-house custom conjugations using the MaxPar X8 antibody-labelling kit (Standard BioTools) following the manufacturers protocol. In addition to lanthanide metals, Indium-115 (Sigma Aldrich) and Platinum-198 (Fluidigm) were used to label antibodies.

Briefly, X8 polymer stored at −20 °C was thawed, resuspended in L buffer and then loaded with 50 mM of lanthanide metal (or In115) at 37 °C for 40 min. Metal loaded polymers were washed twice, firstly with L buffer and 25 min centrifugation, and then with C buffer in a 30 min centrifugation step. During the polymer wash steps 100 µg of purified antibodies were washed with R buffer using a 50 kDa centrifugal unit. Antibodies were then partially reduced with 4 mM TCEP (Fisher) for 30 min at 37 °C. Reduced antibodies were twice washed in C buffer. Partially reduced antibodies were mixed with metal-loaded polymer and incubated at 37 °C for 90 min. Conjugated antibodies were washed and centrifuged four times using W buffer. Purified labelled antibodies were finally eluted from the 50 kDa units by a centrifugation step using 100 µL of W buffer and assessed for protein concentration using a NanoDrop spectrophotometer (ThermoFisher). The antibody preparations were returned to the 50 kDa units for a final buffer exchange step with 100 µl PBS antibody stabilization buffer (Candor). For Pt198 labelling we followed the Maecker lab protocol[59] where platinum directly labels the reduced antibody without the use of polymer. All antibodies were tested at different titres to ascertain the optimal final dilution as follows (format: metal/marker/volume (µl/test)): 89Y/CD45/1.0, 106 Cd/Barcode/0.75, 110 Cd/Barcode/0.75, 111 Cd/Barcode/0.75, 112 Cd/Barcode/0.75, 113 Cd/Barcode/0.75, 114 Cd/Barcode/0.75, 115In/Barcode/0.75, 116 Cd/Barcode/0.75, 148Nd/CD34/0.4, 149Sm/p4E-BP1/0.75, 150Nd/pSTAT5/0.5, 153Eu/pSTAT1/0.5, 156Gd/p38/0.5, 158Gd/pSTAT3/0.5, 159Tb/p-cJun/1, 164Dy/IkBalpha/0.5, 165Ho/CD117/0.75, 166Er/NFkB.p65/0.6, 167Er/CD38/0.5, 172Yb/ki67/0.75, 173Yb/p-Jnk1/Jnk2/1, 175Lu/pS6/0.5, 176Yb/pCREB/0.4, 198Pt/Barcode/0.75, 103Rh/DNA/500 µM, 194Pt/LIVE/DEAD

**CyTOF experimental workflow.** Primary bone marrow-derived white blood cells were sorted for CD34 positivity using a CD34 MicroBead Kit (Miltenyi Biotec) and cultured for 10 days as detailed above (primary cultures) such that cells were actively proliferating. Cells were taken and resuspended to $20-30\times10^6$/ml. Antibody cocktail was prepared in excess and filtered through a 0.1 μm centrifugal filter column (Merck Millipore) to remove antibody aggregates.

Samples were initially barcoded by staining cells with metal labelled CD298/B2M antibodies for 20 min at room temperature (RT). Samples were washed twice with MACS buffer. Resuspended cells were then pooled into a single tube and incubated with Tru-Stain Fc blocking solution (Biolegend) for 10 min at RT. This was immediately followed by incubation with the surface marker antibody cocktail. Staining was performed at RT for 30 min with gentle agitation every 10 min. During the last 2 min of the 30 min incubation, cells were incubated with Cell ID Cisplatin-194 (Pt194). The Pt194 was then quenched with 3 mL MACS buffer. Cells were centrifuged and resuspended in freshly prepared 1.6% paraformaldehyde (Thermo Fisher) and incubated in the dark for 15 min at RT. Cells were washed in MACS buffer then pelleted cells held on ice for 15 min. After a further gentle agitation to ensure cells were well dispersed, 1 mL of cold methanol was added to each tube. Cells were incubated at −20 °C overnight. The next day tubes were allowed to reach RT then washed twice with MACS buffer. Cells were incubated with antibodies for intracellular targets for 30 min at RT. Cells were washed with MACS buffer then stained with 500 μM Rh103 DNA intercalator diluted 1:2000 in 500 ul Fix and Perm buffer (Standard BioTools) at 4 °C overnight.

Samples were acquired within 72 h of cell staining. Prior to acquisition, the samples were washed once with MACS buffer and then twice with freshly dispensed milliQ deionized distilled water (ddH$_2$O). Cells were then resuspended in ddH$_2$O containing 1/10 diluted four element (EQ) normalization beads (Standard BioTools) and filtered through a cell strainer cap (Thermo Fisher). Cell densities were corrected to be lower than $1\times10^6$ cells/ml. Samples were then acquired on a Helios mass cytometer (Standard BioTools) at flow rate of 30 μl/min using a standardized acquisition template following routine tuning and instrument optimization using the HT Helios injector. To ensure absence of sample carryover to the next sample, tubes with milliQ ddH$_2$O (3 min), then wash (nitric acid) solution (3 min) and again miliQ ddH$_2$O (5 min) were run on the instrument in between each sample.

Raw fcs datafiles were (EQ-)bead-normalized using the processing tool in the Fluidigm CyTOF acquisition software. Normalized fcs datafiles were then exported and uploaded to Cytobank software (Beckman Coulter). Each file was cleaned up by a series of manually set gates to exclude normalization beads, non-cellular debris, doublets and dead cells. The processed data was exported into a new experiment where debarcoding was performed to generate individual sample fcs files for further analysis. Processed datafiles were analysed using manual gating using CD45/CD34/CD117 to focus on bulk myeloid cells, then further gated for CD38+/− to focus on LSCs or blasts. Mean ion count data for each channel was exported after confirming normal distribution using biaxial plots and visualised using heatmaps in R. FCS files of gated cells were exported and read into FlowCore v 2.10.0 in R, ion counts were log$_2$ transformed and a pseudocount of 1 added, then a Student's $t$-test performed.

**LSC proliferation assay.** Blood from patient 2 underwent lymphoprep and the cells were sorted using the strategy above for LSCs and blasts. Each population were divided into two, and the membranes stained with (1) PKH-26 (Merck) and (2) CellVue Claret (Merck). The PKH-26 blasts were combined back with the claret LSCs and vice versa, maintaining the original blast:LSC ratio. These cells were then again divided into two and incubated for 6 days in SFEMII media as described above (without hMSCs to avoid contamination), with 20 μM EdU, and with or without 50 ng/ml VEGF and 10 ng/ml IL-5. After 6 days the cells were stained for EdU with the EdU proliferation kit iFluor 488 (Abcam) and flow cytometry was carried out using a CytoFlex (Beckman Coulter). Cells were gated for viability using forward/side scatter, then LSCs/Blasts using PKH-26 (PE) vs Claret (APC) and finally EdU positive/negative (FITC). Gating for PKH-26 and Claret was set using cells which were stained in a known proportion of 90:10 PKH-26:Claret and 10:90 PKH-26:Claret.

**Immunofluorescence.** Cells were adhered to microscope slides using a Shandon Cytospin 4 (Thermo Fisher) at 800 rpm for 3 min. A border was drawn using a PAP pen and cells were then fixed with 4% formaldehyde for 10 min. Permeabilisation was with PBS/0.1% Triton-X100 for 20 min, blocking with PBS/0.1% Tween-20/3% BSA for 1 h. Mouse anti-FLAG antibody (Cat# F3165 Merck) was incubated at 1:100 in PBS/0.1 Tween-20/1% BSA for 1 h, room temperature. Alexa fluor 594 goat anti-mouse antibody (Cat# 115-585-062 Jackson) was incubated at 1:200 in PBS/0.1 Tween-20/1% BSA for 1 h, room temperature. Slides were mounted using ProLong Gold antifade with DAPI (Invitrogen) then imaged using a Zeiss LSM780 confocal microscope, using a Plan Achromat 40 × 1.2 NA water immersion objective, Lasos 30 mW Diode 405 nm and Lasos 2 mW HeNe 594 laser lines. Images were acquired using Zen black version 2.1 and post-acquisition brightness and contrast adjustment was performed uniformly across the entire image.

**In vivo experiments.** All mouse studies were carried out in accordance with UK Animals (Scientific Procedures) Act, 1986 under project licence P74687DB5 following approval from Newcastle University animal ethical review body (AWERB). Mice were housed in specific pathogen free conditions in individually ventilated cages with sterile bedding, water and diet (Irradiated RM3 breeding diet, SDS); with a light/dark cycle of 12 h, relative humidity 45–65% and temperature 20–24 °C. All procedures were performed aseptically in a laminar flow hood. NSG mice (NOD.Cg-Prkdcscid Il2rg tm1Wjl/SzJ) aged between 12 and 16 weeks, both sexes, from an in-house colony were transplanted intra-femorally under isoflurane anaesthetic and 5 mg/kg subcutaneous NSAID analgesia (Carprofen). Newborn MISTRG mice were injected intra-hepatically according to Ellegast et al.[60]. Mice were checked daily, weighed and examined at least once weekly to ensure good health. Endpoints for humane killing were pale extremities, hunched posture, 20% weight loss compared to highest previous weight or 10% weight loss for 3 consecutive days and tumours of 1.5 cm diameter and these criteria were not exceeded in this study.

**Generation of t(8;21) PDX.** Frozen bone marrow cells from relapsed patient #5 were transplanted either intrahepatically or intrafemorally as shown in Table 2. PDX cells were harvested from leg and hip bone BM by clearing the bones of all tissue, crushing and washing in PBS to releash the BM. Spleen blasts were isolated by passing through a 50 μM cell sieve. Cells were washed and stored frozen in 10%DMSO/90%FBS. Peripheral blood blasts were sampled from the tail vein (<10% total blood volume/bleed) and analysed by flow cytometry. Leukemia-inducing cell frequency was calculated by intrafemoral secondary transplantion of PDX isolated from NSG bone marrow, with time to endpoint recorded. PDX is available from the authors on request.

**In vivo inhibition of VEGFA and IL5RA in t(8;21) PDX mice.** Male NSG mice aged between 12 and 16 weeks, were each transplanted intra-femorally (as above) with $0.6\times10^6$ cells from t(8;21) patient #5 secondary transplanted PDX BM. On day 3 after transplant mice were randomised into treatment groups for intra-peritoneal (i.p.) injection (volume < 6 ml/kg, 29 G U100 insulin syringe with needle) of control-vehicle - saline (0.9% NaCl$_2$) $n = 5$ first experiment, $n = 5$ s experiment; Bevacizumab 2 mg/kg in saline $n = 4$ first experiment, $n = 5$ s experiment and Benralizumab 0.38 mg/kg in saline $n = 4$ first experiment, $n = 5$ s experiment. Dosing was continued twice weekly for 13 doses/

**Table 2 | Creation of t(8;21) patient #5 relapse xenograft**

| Transplant (tx) route | Number of cells/mouse in 20 µl | Mouse strain/age | Number of mice engrafted | Latency (weeks) | Sites of engraftment |
|---|---|---|---|---|---|
| intra-hepatic (1.2 Gy 24 h prior to tx) | 1×10E6 | MISTRG/3 days old | 1/5 | 44 | Abdominal tumour |
| intra-femoral | 2×10E6 | NSG/8-12 weeks old | 2/2 | 21, 30 | Bone marrow, spleen (weight 0.1 g), skull, thymus, liver, ascites, ovary, uterus, abdominal tumours, peripheral blood |
| intra-femoral | 2×10E6 | MISTRG/14 weeks old | 1/1 | 32 | Bone marrow, spleen (weight 0.1 g), lung tumour, ovary, uterus, leg tumour, peripheral blood |

mouse. Mice were humanely killed when they reached the endpoints specified above or at 92 days (first experiment) or 99 days (second experiment). One control-vehicle mouse was excluded from analysis as no human cells were detected in blood or bone marrow at harvest, bone marrow from mice harvested due to leg tumour at the injection site was not analyzed due to contamination by the tumour cells. Male mice only were used as (1) the PDX engrafts in the ovaries of females resulting in highly variable latency end points and often before significant bone marrow engraftment and, (2) the bone marrow volume of males is larger than females so more cells could be harvested for analysis.

**RNA isolation, cDNA synthesis and qRT-PCR.** RNA was isolated from Kasumi-1 cells after 2 days after shRUNX1::ETO knockdown was induced with doxycycline using the Nucleospin RNA kit (Macherey-Nagel). cDNA was synthesised using Superscript II (Invitrogen) from 1 µg total RNA, using oligo(dT)12-18 primer. qRT-PCR was carried out using diluted cDNA, Sybr Green PCR Master Mix (Thermo Fisher), 5 µM of sense and antisense primer. Primer sequences as follows: GAPDH sense CCTGGCCAAGGTCATCCAT, antisense AGGGGCCATCCACAGT CTT, RUNX1::ETO sense TCAAAATCACAGTGGATGGGC, antisense CAGCCTAGATTGCGTCTTCACA, VEGFA sense TGCAGATTATGCGGA TCAAACC, antisense

TGCATTCACATTTGTTGTGCTGTAC, ATF4 sense AAACCTCATG GGTTCTCCAG, antisense GGCATGGTTTCCAGGTCATC, FOS sense CGGCCGGGGATAGCCTCTCT, antisense CGGCCAGGTCCGTGCAGA AG, FOSB sense TTGACAATTCTGGGTGCGAGT, antisense CTAAAAGG AAGCCAGGCAATGG, JUN sense TGCTTACCAAAGGATAGTGCGATC, antisense TTGACTTCTCAGTGGGCTTCC, JUND sense TTGACGTGGC TGAGGACTTT, antisense CGCCTGGAAGAGAAAGTGAA, JUNB sense CACCTGCCGTTTACACCAAC, antisense GGAGGTAGCTGATGGT GGTC.

**RNA-seq.** RNA was isolated using the Nucleospin RNA kit (Macherey-Nagel) for Kasumi-1, or RNeasy Plus micro kit (Qiagen) for patient/PDX cells. RNA libraries were generated using TruSeq stranded total RNA library prep kit with ribo-zero for Kasumi-1, or NEBNext Ultra II Directional RNA Library prep kit (New England Biolabs) for primary cells, per the manufacturer's instructions. Illumina sequencing was performed on a NextSeq 550 run in paired-end mode for 150 cycles.

**scRNA-seq.** Patient cells were sorted for LSCs and blasts as described above, then for t(8;21) patients 1 and 2 re-combined at a 1:1 LSC:blast ratio, with 30000 total cells in 45 µl. Cell viability was confirmed then loaded on a Chromium Single Cell Instrument (10X Genomics), to recover 5000 single cells. For patients 3 and 4, LSCs and blasts were loaded on the Chromium controller separately aiming to retrieve as many cells as possible. Library generation for patients 1–3 was performed by the Genomics Birmingham sequencing facility and for patient 4 according to the manufacturer's instructions using the Chromium single cell 3′ library and gel bead kit v3.1. Illumina sequencing was performed on a NovaSeq 6000 S1/NextSeq 500/550 run in paired-end mode for 150 cycles at a depth of 20000 reads per cell.

**DNaseI-seq.** DNaseI digestions were performed as in Bert et al.[61]. Cells were permeabilized in DNaseI resuspension buffer (60 mM KCl, 10 mM Tris pH7.4, 15 mM NaCl, 5 mM MgCl2 and 300 mM sucrose) and then DNaseI diluted in dilution buffer (60 mM KCl, 0.4% NP40, 15 mM NaCl, 5 mM MgCl2, 10 mM Tris pH 7.4 and 2 mM CaCl2 was added and incubated at 22 °C for exactly 3 min. The digestion was terminated by adding cell lysis buffer (300 mM Sodium Acetate, 10 mM EDTA pH 7.4, 1% SDS and 1 mg/ml proteinase K). DNA was purified using phenol-chloroform extraction. Library preparation was performed using the KAPA HyperPrep kit (Roche) on extracted DNA with size selection for

200–300 bp fragments and sequenced on a NextSeq 550 (Illumina) run in single-end mode for 75 cycles.

**ATAC-seq.** Omni ATAC-seq was performed as in Corces et al.[62]. Briefly, cells were washed in ATAC resuspension buffer (RSB) (10 mM Tris-HCl pH7.5, 10 mM NaCl and 3 mM MgCl2) and then lysed for 3 min on ice in RSB buffer with 0.1% NP-40, 0.1% Tween-20. Then the cells were washed with 1 ml of ATAC wash buffer consisting of RSB with 0.1% Tween-20. Nuclei were resuspended in ATAC transposition buffer consisting of 25 µl TD buffer and a concentration of Tn5 transposase enzyme (Illumina) related to the number of input cells up to 2.5 µl, 16.5 µl PBS, 5 µl water, 0.1% tween-20 and 0.01% digitonin and then incubated on a thermomixer at 37 °C for 30 min. The transposed DNA was then amplified by PCR amplification up to ¼ of maximum amplification, as assessed by a qPCR side reaction and sequenced on a NextSeq 550 (Illumina) run in single-end mode for 75 cycles.

**ChIP-seq.** Between 2 and $20 \times 10^6$ cells (number is antibody dependent) were crosslinked following 72 h of dnFOS induction with doxycycline, using 1% formaldehyde for 10 min at room temperature. For GATA2 and FOS cells were double crosslinked, by adding 415 µg/ml Di(N-succinimidyl) glutarate for 45 min prior to formaldehyde crosslinking. Cells were lysed and nuclei extracted using lysis buffer (10 mM HEPES pH 8.0, 10 mM EDTA pH 8.0, 0.5 mM EGTA pH 8.0, 0.25% Triton X-100, protease inhibitor cocktail (PIC) 1:100) followed by nuclear lysis buffer (10 mM HEPES pH 8.0, 1 mM EDTA pH 8.0, 0.5 mM EGTA pH 8.0, 0.01% Triton X-100, 200 mM NaCl, PIC 1:100). Nuclei were sheared to around 100–600 bp in sonication buffer (25 mM Tris pH 8.0, 150 mM NaCl, 2 mM EDTA pH 8.0, 1% Triton X-100, 0.25% SDS, PIC 1:100), using a Picoruptor (Diagenode) for between 4 and 16 cycles of 30 s on/30 s off (cycle number dependent on cell number and crosslinking). Sheared chromatin was diluted in IP buffer (25 mM Tris 1 M pH 8.0, 150 mM NaCl, 2 mM EDTA pH 8.0, 1% Triton X-100, 7.5% Glycerol, PIC 1:1000). Dynabeads protein G were pre-incubated with antibodies against FOS (Cat# MA5-15055 ThermoFisher), CEBPA (Cat# sc-61X Santa Cruz), RUNX1 (Cat# ab23980 Abcam), RUNX1::ETO (Cat# C15310197 Diagenode), PU.1 (Cat# sc-352 Santa Cruz), GATA2 (Cat# AF2046 R & D Systems), H3K27ac (Cat# ab4729 Abcam), H3K9acS10P (Cat# ab12181 Abcam) or H3K4me3 (Cat# 04-745 Millipore) for 2 h at 4°, then added to the chromatin. Chromatin and antibody-beads mixture were incubated for between 4 and 18 h (antibody dependent) at 4°. Beads were then washed sequentially: once with buffer 1 (20 mM Tris pH 8.0, 150 mM NaCl, 2 mM EDTA pH 8.0, 1% Triton X-100, 0.1% SDS), twice with buffer 2 (20 mM Tris pH 8.0, 500 mM NaCl, 2 mM EDTA pH 8.0, 1% Triton X-100, 0.1% SDS), once with buffer 3 (10 mM Tris pH 8.0, 250 mM LiCl, EDTA pH 8.0, 0.5% NP-40, 0.5% Sodium deoxycholate), twice with buffer 4 (10 mM Tris pH 8.0, 50 mM NaCl, 1 mM EDTA pH 8.0). Enriched DNA was eluted from the beads with 100 mM sodium bicarbonate and 1% SDS. Crosslinks were reversed with 25 µg Proteinase K for 16 h at 65 °C and DNA was purified using AmpureXP beads (Beckman Coulter). Enrichment was confirmed using qRT-PCR with known positive and negative binding sites for each protein target, then library preparation and sequencing was carried out as for DNaseI-seq with size selection for 200–500 bp fragments.

**CUT&RUN.** Nuclear CUT&RUN was performed as in Skene and Henikoff [63]. Briefly, $1 \times 10^5$ cells were washed with PBS. Nuclei were isolated with NE Buffer (20 mM Hepes-KOH pH 7.9, 10 mM KCl, 0.5 mM spermidine, 0.1% Triton X-100, 20% Glycerol), captured with Concanavalin A beads (Bangs Laboratories, BP531) and incubated with anti-H3K27me3 antibody (Cat# 9733 Cell signalling) for 2 h at 4 °C. After washing away unbound antibody with wash buffer (20 mM HEPES-NaOH pH 7.4, 150 mM NaCl, 0.5 mM Spermidine, 0.1% BSA and 1x protease inhibitor cocktails from Sigma), protein A-MNase (provided

by the Henikoff laboratory) was added at a 1:200 ratio and incubated for 1 h at 4 °C. The nuclei were washed again and were equilibrated to 0 °C on a metal block and MNase digestion was activated with CaCl2 at a final concentration of 2 mM for 5 min. The digestion was terminated with the addition of equal volume of 2xSTOP buffer (200 mM NaCl, 20 mM EDTA, 4 mM EGTA, 50 mg/mL RNase A and 40 mg/mL glycogen). The protein-DNA complex was released by centrifugation and then digested by proteinase K at 70 °C for 10 min and DNA was purified using phenol-chloroform extraction. Library preparation was performed using the KAPA HyperPrep kit (Roche) on extracted DNA and sequenced on a NextSeq 550 (Illumina) run in single-end mode for 75 cycles.

**RNA-seq analysis.** Raw paired-end reads were processed with Trimmomatic v0.39 [64] to remove sequencing adaptors and low-quality sequences. The processes reads were then aligned to the human genome (version hg38 https://www.ensembl.org/info/data/ftp/index.html) using Hisat2 v2.2.1 [65] with default parameters.

Gene expression from sorted LSC and blast experiments were calculated as fragments per kilobase of transcript per million mapped reads (FPKM) using Stringtie v2.1.3 [66] with default parameters and gene models from Ensembl as the reference transcriptome. Only protein-coding genes that were expressed with an FPKM value > 1 in either the LSC or blast samples were retained for further analysis. FPKM values were normalized using upper-quartile normalization and further log2-transformed with a pseudocount of 1 added before transformation. A gene was considered to be either LSC or blast specific if it had a fold-change > 1 between cell types.

Counts from all other RNA-Seq experiments were obtained using featureCounts [67] from the Subread package v2.0.1 using the options -p -B -s2 and gene models from refSeq as the reference transcriptome. Only genes with at least 50 counts in at least one sample were retained for further analysis. Counts were normalized using the edgeR package [68] in R v4.1.0, and differential gene expression analysis was then carried out using limma-voom [69]. For experiments where replicates were available, a gene was considered to be differentially expressed if it had a fold-change of at least 2 and a Benjamini-Hochberg adjusted $p$-value < 0.1. In cases where no replicates were possible, only a 2-fold-change was used.

Gene set enrichment analysis (GSEA) was carried out using the GSEA software (Broad Institute) [70]. Genes were ranked by the $\log_2$ fold change and a normalised enrichment score and nominal p-value were calculated for gene sets comprising the LSC or Blast specific differential genes.

Published processed data was obtained from GSE108316 [7].

**ATAC/DNaseI-seq analysis.** ATAC or DNaseI sequencing reads were processed with Trimmomatic v0.39 to remove sequencing adaptors and low-quality reads. Trimmed reads were aligned to the human genome (version hg38) using Bowtie2 v2.3.5.1 [71] using the setting --very-sensitive-local. PCR duplicates were removed using the MarkDuplicates function in Picard 2.21.1. Bigwig files were made using the bamCoverage function in deepTools 3.5.0 [72] and normalised as counts per million (CPM). These bigwig files were then plotted using the UCSC genome browser. Peaks were called using MACS2 v2.2.7.1 [73] using the settings -q 0.0005 -B --trackline --nomodel --shift -100 --extsize 200.

To carry out differential chromatin accessibility analysis, a peak union was generated using the bedtools v2.29.2 [74] merge function. The average tag-density in a 400-bp window centred on the peak union summits was calculated for each sample using the annotatePeaks.pl function in Homer v4.11 [75] using the bedGraph files generated by MACS2. These were then normalised as CPM and further log2-transformed as $\log_2(CPM + 0.1)$. Peaks were considered to be differentially accessible if there was at least a 2-fold difference between samples.

Density plots were generated using Homer v4.11 annotatePeaks.pl function using the bedGraph files generated by MACS2, with the options -size 2000 -hist 10 -ghist and plotted using JavaTreeView 1.1.6.

In order to measure if a transcription factor motif was over-represented in a set of differentially accessible peaks, we calculated a motif enrichment score (ES) as follows. The number of motifs in a peak set was first counted by extracting the motif positions using the find-MotifsGenome.pl function in Homer with the options -size 200 -find. The probability weight matrices provided by the Homer motif database were used in all analyses. The enrichment score was then calculated as Eq. 1:

$$S_{ij} = \frac{n_{ij}/m_j}{\sum_j n_{ij}/\sum_j m_j} \qquad (1)$$

where i is the motif, j is the peak set, $n_{ij}$ is the number of sites in peak set j that contain the motif i and $m_j$ is the total number of sites in peak set j. The scores were then hierarchically clustered using complete linkage of the Euclidean distance in R and displayed as a heat map.

Average profiles were created using normalized bigwig files. To do this, the average peak height for each sample was calculated for each sample using the computeMatrix and plotProfile functions in deep-Tools. A normalization factor was then calculated for each sample so that the average peak height was the same for all samples. Normalized bigwig files were then created using the bamCoverage function in deepTools using the --scale option to apply the normalization factor. The average profile was then plotted using the computeMatrix and plotProfile functions.

Average motif profiles were generated using Homer annotate-Peaks.pl with the options -size 2000 -hist 10 -m <target motif position weight matrices> and plotted using R ggplot2 using the geom_smooth loess function.

Published raw data was obtained from GSE108316[7] and GSE211095[10] and processed as above.

**ChIP-seq/CUT&RUN analysis.** ChIP-sequencing reads were processed with Trimmomatic v0.39 to remove sequencing adaptors and low quality reads. Trimmed reads were aligned to the human genome (version hg38) using Bowtie2 v2.4.4 using the setting --very-sensitive-local. PCR duplicates were removed using the MarkDuplicates function Picard v2.21.1. Bigwig files were created for viewing in UCSC genome browser using deepTools 3.5.0 bamCoverage, with normalisation using counts per million (CPM). Peaks were called using MACS2 using the settings -q 0.01 -B --trackline. Differential peaks were calculated as for ATAC-seq.

Average profiles were generated as for ATAC-seq, except for H3K27me3 where normalisation was only by counts per million due to the broad regions which have this mark. The average peak height was calculated from these profiles at specific sites and a $\log_2$ fold change calculated and plotted as a heatmap in R using hierarchical clustering as for ES above. ES and density plots were generated as for ATAC-seq except for H3K27me3 and H3K4me3 where density plots were generated using deepTools plotHeatmap in conjunction with the average profiles.

In order to ensure that ChIP peaks were associated with the correct target gene we used processed promoter-capture Hi-C data from Assi et al.[7]. This was done by first searching for peaks that could be assigned to a DNaseI hypersensitive site (DHS) for which the Hi-C data could associate the DHS with the correct gene promoter. In cases where no Hi-C association was available, peaks were assigned to their closest gene based on transcription start site (TSS) using the annotatePeaks.pl function in Homer.

Published raw data was downloaded from GSE29225[12] and processed as above.

**scRNA-seq analysis.** Reads from single-cell RNA-Seq experiments were aligned to the human genome (version hg38) and quantified using the count function in CellRanger v6.0.1 from 10x Genomics and using gene models from Ensembl as the reference transcriptome. Unique Molecular Identifier (UMI) count data was filtered for low quality cells by removing cells with less than 200 and more than 5000 detectable genes. Cells that had more than 15% of UMIs aligned to mitochondrial transcripts were also excluded from further analysis. UMI counts were normalized using the log-normalize method in the Seurat package v4.3.0[76] in R v4.1.2. The cell cycle stage was then estimated for each cell using the CellCycleScoring function in Seurat and using the in-built lists of cell cycle stage associated genes. To account for the possible effect of cell cycle stage on downstream clustering analysis, S-phase and G2M-phase scores were included as variables in a linear regression model using the ScaleData function in Seurat. Principal Components Analysis (PCA) was then performed on the normalized and scaled data, with the number of principal components selected per sample for further analysis. Cells were then clustered using the FindClusters function in Seurat with a resolution value of 0.8 and visualized using Uniform Manifold Approximation and Projection (UMAP). Cluster marker genes, corresponding to genes that are significantly higher expressed in a cluster compared to all other cells outside of that cluster, were identified using the FindAllMarkers function. Genes that had an average log2-fold change of at least 0.25 with an adjusted p-value less than 0.1 were selected as marker genes. For t(8;21) #3 and #4 this process was carried out on each LSC and Blast library separately, they were then integrated by using the functions SelectIntegrationFeatures to identify the anchor features, followed by FindIntegrationAnchors using the previously identified anchor features and reduction method "rpca" to avoid over-integration. Finally the datasets were integrated with these defined anchors using the function IntegrateData. Clusters with less that 4 cells expressing *RUNX1T1* were determined to be contaminating healthy cells and removed (patient 3 only, sample from peripheral blood).

In order to classify a single-cell cluster as either blast or LSC, for t(8;21) #1 and #2 specific genes from the blast and LSC bulk RNA-seq above were used as a reference gene expression signature for Gene Set Enrichment Analysis (GSEA). GSEA was carried out using the fgsea package v1.10.1 (27) in R. To do this, cluster marker genes from single-cell clusters were used as pathways and compared to the gene expression signatures derived from the bulk data. This analysis produced a normalised enrichment score (NES) for each cluster, with a positive NES suggesting that a cluster has a more blast-like gene expression signature and a negative NES suggesting a more LSC-like signature. Only clusters with a Benjamini-Hochberg adjusted p-value < 0.05 and an absolute NES > 1 were considered to be positively classified as either LSC or blast, with intermediate LSC/Blast clusters defined as those not positively classified as LSC or blast. For t(8;21) #3 and #4, the identity determined by sorting was plotted and the cluster defined as LSC or blast based on the majority component, or as intermediate LSC/Blast where a cluster was mixed.

All 4 patient datasets were then integrated as above but using the default reduction and merged using the merge function in Seurat. The integrated dataset was then taken forward for further analysis, rescaled and processed as above to find new clusters, with all data plotted on the UMAP generated from the integrated dataset. Clusters were again labelled as LSC, Blast or LSC/Blast based on the contributing cells from the individual patients. LSC and Blast marker genes were then identified using the FindAllMarkers function, with all genes with an average log2 fold change of 0.5 (positively enriched only) and adjusted p-value < 0.1 taken.

Single-cell trajectory analysis was carried out using Monocle3 v1.3.4[77]. Processed data from Seurat was imported to Monocle and trajectories were inferred using the learn_graph function. Pseudotime was then calculated using the order_cells command, using cells from

the earliest inferred LSC population as the root. Trajectories were then plotted on the UMAP calculated by Seurat.

$Z$-scores of t(8;21)-specific genes were calculated by first calculating the average gene expression per cluster using the AverageExpression function in Seurat. The t(8;21)-specific genes were calculated using normalised FPKM values from bulk AML samples obtained from Assi et al.[7], with genes considered as t(8;21)-specific if they were at least 2-fold higher in the average of all t(8;21) patients compared to the average of each of the other AML subtypes or PBSCs. The average cluster expression of the t(8;21)-specific set of genes was then transformed to a $Z$-score using the scale function in R and plotted as a heatmap with supervised clustering by cell cluster ordered by the inferred pseudotime trajectory and ordered from highest to lowest $Z$ score in each population.

Genes that were specifically differential in $G_0/G_1$ cells were obtained by subsetting all of the non-S/G2M phase cells based on the cell cycle scoring above. The FindAllMarkers function was then run on this subset using the LSC/Blast classification rather than the clusters. All genes were then used for GO term analysis. Gene ontology (GO) term analysis was carried out using DAVID 6.8. GO terms present in at least 3 patients selected for further analysis. GO term results were then visualised as a bubble-plot in R v4.1.0 with the size of each bubble representing the adjusted $p$-value, and the colour corresponding to the percentage of genes from that GO term that were present in the set of differentially expressed genes.

Healthy cell data was downloaded from the Human Cell Atlas https://explore.data.humancellatlas.org/projects/455b46e6-d8ea-4611-861e-de720a562ada as an h5ad file and loaded into Seurat. Analysis was carried out in the same manner as for our AML cells, with the top HSC and MP cluster identified using the authors' marker genes[25] prior to subsetting based on cell cycle stage to find marker genes.

**Statistics and reproducibility.** For comparisons of in vitro drug/cytokine treatment vs control only two-sided Student's t-tests or one way ANOVA with Bonferroni's multiple testing were performed as appropriate to the number of conditions being compared. In vivo flow cytometry data was analysed using Student's $t$-tests with Welch's correction. For growth curves two-way ANOVA was performed with Dunnett correction for multiple comparisons at each time point. For mass cytometry data Student's $t$-tests were performed on $\log_2$ transformed data. Sample size for in vivo experiments was based on technical limitations, power calculation suggests between 13 and 23 mice would be required for the effect size observed; data were excluded based on contaminating tumour cells as written in the method for this experiment. For in vitro experiments no statistical method was used to predetermine the sample size and no data were excluded from the analyses. Experiments were not randomized and investigators were not blinded to allocation during experiments and outcome assessment.

### Reporting summary

Further information on research design is available in the Nature Portfolio Reporting Summary linked to this article.

## Data availability

RNA-seq, scRNA-seq, ATAC-seq, DNaseI-seq and ChIP seq data generated in this study have been deposited in the Gene Expression Omnibus (GEO) under accession code GSE226603. Data from cell growth assays, gate percentages from flow cytometry and qPCR data are provided in the Source Data file. Published processed data was obtained from GSE108316[7] and the Human Cell Atlas https://explore.data.humancellatlas.org/projects/455b46e6-d8ea-4611-861e-de720a562ada. Published raw data was obtained from GSE108316[7], GSE211095[10] and GSE29225[12]. Human genome hg38 was downloaded from Ensembl https://www.ensembl.org/info/data/ftp/index.html Source data are provided with this paper.

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

## Acknowledgements

This work was funded by grants to C.B. and P.N.C. from Blood Cancer UK (20006), a grant from the Medical Research Council (MR/S021469/1) to C.B., P.N.C and O.H, a Cancer Research UK programme grant (C27943/A23389) and a KIKA programme grant (329) to OH, a Leukemia UK John Goldman Fellowship (2021/JGF/001) to D.J.L.C and a Cancer Research UK studentship to A.A. Benralizumab was obtained from Astra Zeneca in the context of the Open Innovation scheme. We would like to thank the West Midlands Regional Genetics Laboratory for supplying mutation data linked to AML patient samples. The authors would like to acknowledge Celina Whalley of Genomics Birmingham and Guillaume Desanti of the University of Birmingham Flow Cytometry facility for support of next-generation sequencing and cell sorting experiments. We would also like to thank Hesta McNeill and Samantha Jepson Gosling of Newcastle University, and Mauricio Ferrao Blanco, Alicia Perzolli, Elizabeth Schweighart, Bexultan Kazybay, Aleksandra Balwierz and Philip Lijnzaad from the Princess Maxima Center for their technical assistance. Finally, thanks to Ellen Rothenberg for an illuminating conversation regarding the *KIT/KDR* shared enhancer.

## Author contributions

S.G.K. and C.B. conceived and directed the work and wrote the paper, S.G.K., S.P., H.J.B., P.S.C, P.K.D. A.P., A.W., L.A. A.A., D.J.L.C., N.K., A.K.-H. performed experiments, M.R. and S.P. provided patient cells, S.G.K., P.K and S.A.A. analysed data, P.N.C. helped writing the paper, H.J.B. and O.H. directed the mouse work and helped writing the paper.

## Competing interests

The authors declare no competing interests.
