## [Peer Review File · Nature Communications]

REVIEWER COMMENTS

Reviewer #1 (Remarks to the Author); expert in AML and LSC:

Summary: Kellaway et al. start by comparing chromatin conformation and transcriptomes (bulk & single cell) of the CD34+/CD38- putative leukemic stem cells (LSC) fraction of bone marrow samples of two t(8;21)+ AML patients. They found differential expression of the fusion as well as classical t(8;21) genes in the LSC vs. the bulk compartment including IL5RA and VEGFA that were both higher in the LSC than the bulk (Fig.1). They defined cell cycle-associated differentially expressed genes and phosphorylated proteins in LSC vs. bulk cells with IL5RA and VEGFA being LSC-enriched (Fig.2). Functional experiments in two t(8;21)+ cell lines (KASUMI-1 & SKNO-1) indicated that VEGF and IL5 signaling promote growth of these cells; in addition, IL5 as well as VEGF stimulated growth of primary LSC and bulk AML cells from one patient (Fig.3). PDX experiments in NSG mice (with cells from another t(8;21)+ patient after relapse) showed that blocking IL5 or VEGF signaling impaired leukemia-initiation: notably LSC showed higher IL5RA expression (Fig.4). Mechanistic studies in KASUMI-1 and SKNO-1 cells conditionally expressing a dominant-negative (dn-) acting FOS protein indicated that IL5-and VEGF growth signals converge at AP-1 (Fig.5). Multi-omics analysis of KASUMI-1 clone expressing dnFOS and/or dnCEBP suggested that interference with AP-1 silences genes in leukemic blasts by shifting FOS and PU.1 to GATA2 sites a loss of RUNX1 from AP-1 and C/EBP sites (Fig.6). In addition, reactivation of LSC-signature genes by AP-1 inhibition was shown in primary t(8;21)+ AML cells (Fig.7). Finally, experiments using a conditional shRNA targeting the driving RUNX1-ETO fusion in KASUMI1 cells revealed that the fusion controls AP-1 expression and localization that is further regulated by IL5 and VEGF signals (Fig.8).

Overall, this is a very-well performed study with a MS very rich in data. The paper is well-written and the figures are of high quality. Nevertheless, the reviewer feels that there are some conceptual limitations that may confuse not only the reviewer but maybe also other readers.

First, and most importantly, the study starts with identification of LSC-selective aberration which leads to the identification of activation of IL5 and VEGF signals. However, the majority of the functional studies are done in KASUMI-1 (and some also in SKNO-1) cell lines. The reviewer understands that these are the well-established models for molecular assays of t(8;21)+ AML that need a lot of cells, but do they really reflect LSC biology? It seems that one starts with LSC then moves to cell lines, and in the end to generally valid concepts for t(8;21)+ AML? Although nicely performed, the majority of the functional studies have been done in KASUMI1 cells and they are neither primary AML cells nor LSC. The authors should at least discuss these limitations which do not reduce the merits of their efforts, but helps to better understand the work but also not to conceptually confuse any reader.

In addition, it is also rather confusing that some experiments are done with patients 1&2, then a third patient appears and finally some in vivo work is done with cells from a fourth patient: even though they are all t(8;21)+ it seems that they were selected to provide the most suitable results and/or sufficient material to perform a particular experiment. This should be clearly stated throughout the MS.

Detailed comments:

1. Lines 115-118: The authors find expression of the AML-ETO transcript in 73 and 58% of cells "across all LSC and blast clusters confirming they were all AML or pre-leukemic". Here the authors should dissect LSC and bulk. The illustration (Fig.1F) suggests a generally higher expression of the fusion in the LSC compartment than the bulk. In addition, there seems to be a continuum from LSC to bulk cells: is this also somehow correlating with the level of fusion mRNA? Could this be quantified on a single cell level? The reviewer however, does not understand the statement that fusion gene expression in a fraction of the cells suggests that the cells are "all AML or pre-leukemic"? Is AML-ETO not seen as a preleukemic initiation event in this type of AML? Why do some cells express low levels (or not at all) the fusion? Can they separate fusion + from fusion - cells and compare biologic behavior?

2. To validate the nature of CD34+/CD38- cells as LSC, the authors compared the clonogenic activity of them with the CD34-/CD38+ bulk cells (Extended data figure 1B). Why was this done

only with cells from patient #2? Did the bulk cells form clusters? The picture of a sole colony is not really informative: one would like to have a broader picture with e.g. type of colonies? It may be difficult if material is sparse, nevertheless the overall heterogeneity of the two patients studied implies a direct comparison.

3. Fig.1G shows a heatmap of hierarchically clustered gene expressed in LSC and bulk clusters from two patients: they marked IL5RA and VEGFA enriched in the LSC compartments of both of the patients. Where other genes significantly differentially expressed in LSC from both of the patients?

4. Fig.2A shows the cell cycle gene expression distribution of cells from both patients. For this reviewer the quantitative figure (extended Fig.2A) is more informative: could the authors at least give some numbers/percentage in the MS text?

5. Fig.2C compares phosphorylation status (CYTOF?) of signaling molecules in CD34+ AML cells from two patients (#2 and #3). This is difficult to understand. First, why now another patient? If sufficient viable material was not available from #1, why was all the epigenomic analysis done with this patient and NOT with patient 3? Or was this analysis also done but did not provide "suitable" data? In addition, the authors claim significant higher phosphorylation of CREB, JUN and JNK1/2 in bulk vs. LSC while e.g. STATs seem not changed: however the numbers in the right part of the panel suggest overall minor changes and in fact increased pSTATs in LSC than bulk. They need to present that in a clearer way that also "on-experts" can follow.

6. Line 142: "...whether at LSC use..." is this sentence correct?

7. Lines 155-157: The reviewer does not understand the statement that activating KIT mutations be insufficient to initiate LSC growth despite being found equally in LSC and blasts? Is this shown here? Previous studies showed that (viral) overexpression of activating KIT mutations cooperate with AML-ETO to induce a leukemic phenotype (<https://doi.org/10.1073/pnas.1019625108>; <https://doi.org/10.1182/blood-2011-02-338228>)? It can well be that some LSC have an activating KIT mutation and are quiescent, as following the stem cell hypothesis, only a minority of LSC are actively cycling? Could it be that the expression level of the fusion determines expression of putative growth regulators: it is interesting that IL5RA and VEGFA seems at least in part to correlate (Figs. 1F and 2D)? How does IL5RA and VEGFA correlate with the G1 phase of the cycle at a single cell level?

8. Experiments shown in Fig.4 show nicely reduction of LSC by blocking IL5RA (Benralizumab) or VEGF (Bevacizumab) in a PDX model of t(8;21)+ AML. One wonders whether such a treatment also prolongates the latency until disease symptoms respectively survival (Kaplan-Meier plot)? Or would it at least translate into significantly altered of the PDX cells into a secondary host?

9. Why does the expression of dnFOS increases the clonogenic activity in the 2nd plate (replating)?

10. Fig7B shows differential gene expression in primary human normal CD34+ HSCP and CD34+ PDX cells upon inducible activation of dnFOS: how many and which genes were deregulated in both or more in the PDX cells? It is difficult to understand that AP-1 seems required for the maintenance of the blast genetic program as initially the authors stressed expression and activation of IL5RA and VEGF leading to AP-1 activation in LSCs? In addition, how does (as shown in the schematic 8F) activation of AP-1 in LSC will induce myeloid transcription factors in the blast cell compartment? Would that mean that AP-1 would shift the LSC towards the blast compartment: in other words, more IL5RA and VEGF-induced AP-1, the smaller the LSC compartment? This is confusing.

Reviewer #2 (Remarks to the Author); expert in leukaemia and VEGF signaling:

In this paper the authors show that patients with acute myeloid leukemia (AML) sub-type t(8;21) harbor leukemic stem cell (LSC) that aberrantly activate VEGF-A and IL-5 signaling pathways.

Incubation of AMLs with VEGF-A and IL5 results in the expansion of LSCs. In a PDX model inhibition of IL5 and VEGF-A signaling diminishes the expansion of the LSCs. Employing ATAC-seq and ChIP analyses carried out on Kasumi cell line, they show that dysregulation of the oncoprotein RUNX1::ETO and activation of AP-1/GATA2 axis augments t(8;21) LSCs self-renewal and progression.

Comments:

Identifying the molecular pathways that are overactive in certain leukemias such as t(8;21) AMLs could lead to development of novel strategies to target leukemic stem cells (LSCs). The finding that t(8;21) in AML results in the aberrant overactivation of VEGF-A and IL5 could lead to designing new approaches to target LSC-driving leukemias.

However, targeting VEGF-A and/or IL5 results in a modest response in leukemic progression. In addition, numerous previous have shown that inhibiting VEGF-A and its various receptors such as VEGFR2 or VEGFR1 only partially induce regression of leukemias. Most importantly, the mechanism by which targeting VEGF-A and IL5 results in LSC regression or decreased self-renewal is unclear.

Specific comments:

- 1) How does t(8;21) in AMLs and activation of RUNX1::ETO and AP-1/GATA2 results in induction of IL5 or VEGF-A. Are IL5 or VEGF-A induced in leukemic cells themselves or in the niche cells?
- 2) Which niche cells are presumably activated by LSCs, and how they are induced to produced to release VEGF-A or IL5?
- 3) Which receptors for VEGF-A (VEGFR1, R2 or R3) and IL5 are expressed on LSCs? Is the expression of these receptors dependent on the factors released specifically by t(8,21) and RUNX1::ETO or AP-1/GATA2 axis?
- 4) Why for the ATAC-seq and ChIP analyses the authors have elected to use Kasumi cell line rather than primary leukemic cells that harbor t(8,21) translocation.
- 5) The authors should consider providing histological analyses of the leukemic bone marrow after targeting of IL5 and VEGF-A.

Reviewer #3 (Remarks to the Author); expert in epigenetics and transcriptional regulation:

In this manuscript, Sophie G Kellaway et al., by using a multidisciplinary approach, tried to dissect regrowing of LSCs, a process commonly associated with relapse induction after therapy. Specifically, the authors by using the t(8;21) AML sub-type as a model of investigation (cells, cell lines and patient-derived xenograft model (PDXs)) claim that VEGF and IL-5 signalling pathways are aberrantly activated in LSCs and are required for LSCs growth. Importantly the authors define that both pathways act via regulatory circuits formed by RUNX1-ETO and an AP-1/GATA2 axis giving rise to the LSCs to re-enter the cell cycle while preserving self-renewal capacity.

Overall, the manuscript draft is detailed and readable representing a potential contribution to the field with an appreciable effort in identifying molecular patterns driving LSCs growth. However, some gaps in the manuscript prevent its publication in the current form and these concerns need to be addressed by the authors.

Major comments

- 1- The authors identified an LSC signature by applying single-cell (sc) approaches in only two

patients. Although the analysis seems to be well performed, the number of patients used is a limitation since it cannot represent a "common" line in LSCs and is undoubtedly not enough for building the roadmap of the paper and defining identified genes as involved in the LSC maintenance. Thus, the authors should implement this data by using more patients (for example they used pt 3 and pt 4 in the paper, why do not include them at least for the most important and informative sc analysis?). Additionally, I found that expression levels of VEGF and IL5RA are quite variable in the two patients, and the correlation between RNA-seq and sc for KDR appears weak suggesting (also in this case) that more samples might be needed and likely of help.

2- To corroborate their findings, authors could compare results obtained from experiments performed in Fig 2 with "normal samples" (if possible) and analyze gene expression in normal blasts, HSC, and leukemia.

3- To deeper confirm the involvement of VEGF and IL5 signalling on t8;21 leukemic cells the authors should carry out additional phenotypic analysis, such as CCK8 assay, cell cycle analysis, or competition assay (etc).

4- According to Fig 4J and F, the results displayed are not significant, can the authors address this point?

5- It is unclear why the authors chose Kasumi-1 as a cell model to investigate molecularly the role of AP1 since this cell line does not harbor IL5R. Can the authors better clarify this point? (Fig 6)

6- Based on the results, on one hand, it seems that targeting AP1 contributes to affecting leukemic phenotype, however, it also induces LSC proliferation via GATA binding. Such a story has only been persecuted molecularly without a phenotypic readout (if present is very poor).

Therefore, could the author asses phenotypically that targeting AP-1 induce a shift from LSC to blast cells? Additionally, authors should better explore (and also comment on) how targeting AP1 implies leukemic cell growth and at the same time induces LSC self-renewal and proliferation.

7- Fig 8. Can authors corroborate and strengthen these findings by performing experiments by knocking down RUNX1-ETO following IL5 and VEGF stimulation?

Minor comments

To confirm the specificity of the pathways (IL5 and VEGF) could authors analyze the effects of these cytokines also in other leukemic models (by using cell lines)?

Figure 4D shows sampling at different time points of blood samples, however, it is not clear how these samples have been used in the results readout.

Stdev needs to be added in FigS1C.

RESPONSE TO REVIEWERS' COMMENTS

Reviewer 1:

Summary: Kellaway et al. start by comparing chromatin conformation and transcriptomes (bulk & single cell) of the CD34+/CD38- putative leukemic stem cells (LSC) fraction of bone marrow samples of two t(8;21)+ AML patients. They found differential expression of the fusion as well as classical t(8;21) genes in the LSC vs. the bulk compartment including IL5RA and VEGFA that were both higher in the LSC than the bulk (Fig.1). They defined cell cycle-associated differentially expressed genes and phosphorylated proteins in LSC vs. bulk cells with IL5RA and VEGFA being LSC-enriched (Fig.2). Functional experiments in two t(8;21)+ cell lines (KASUMI-1 & SKNO-1) indicated that VEGF and IL5 signaling promote growth of these cells; in addition, IL5 as well as VEGF stimulated growth of primary LSC and bulk AML cells from one patient (Fig.3). PDX experiments in NSG mice (with cells from another t(8;21)+ patient after relapse) showed that blocking IL5 or VEGF signaling impaired leukemia-initiation: notably LSC showed higher IL5RA expression (Fig.4). Mechanistic studies in KASUMI-1 and SKNO-1 cells conditionally expressing a dominant-negative (dn-) acting FOS protein indicated that IL5-and VEGF growth signals converge at AP-1 (Fig.5). Multi-omics analysis of KASUMI-1 clone expressing dnFOS and/or dnCEBP suggested that interference with AP-1 silences genes in leukemic blasts by shifting FOS and PU.1 to GATA2 sites a loss of RUNX1 from AP-1 and C/EBP sites (Fig.6). In addition, reactivation of LSC-signature genes by AP-1 inhibition was shown in primary t(8;21)+ AML cells (Fig.7). Finally, experiments using a conditional shRNA targeting the driving RUNX1-ETO fusion in KASUMI1 cells revealed that the fusion controls AP-1 expression and localization that is further regulated by IL5 and VEGF signals (Fig.8).

Overall, this is a very-well performed study with a MS very rich in data. The paper is well-written and the figures are of high quality. Nevertheless, the reviewer feels that there are some conceptual limitations that may confuse not only the reviewer but maybe also other readers. First, and most importantly, the study starts with identification of LSC-selective aberration which leads to the identification of activation of IL5 and VEGF signals. However, the majority of the functional studies are done in KASUMI-1 (and some also in SKNO-1) cell lines. The reviewer understands that these are the well-established models for molecular assays of t(8;21)+ AML that need a lot of cells, but do they really reflect LSC biology? It seems that one starts with LSC then moves to cell lines, and in the end to generally valid concepts for t(8;21)+ AML? Although nicely performed, the majority of the functional studies have been done in KASUMI1 cells and they are neither primary AML cells nor LSC. The authors should at least discuss these limitations which do not reduce the merits of their efforts, but helps to better understand the work but also not to conceptually confuse any reader.

Response: We thank the reviewer for his/her positive comments. We have added clarifying comments addressing this issue where appropriate. As outlined in further detail below, we used primary cells wherever possible and have added additional explanations of the limitations of Kasumi-1 as a model for LSCs. However, please note that we still regard it as significant that the abolition of AP-1 activity brings back GATA2 activity and upregulates a gene expression pattern that we find in primary LSCs. This result makes a lot of biological sense.

In addition, it is also rather confusing that some experiments are done with patients 1&2, then a third patient appears and finally some in vivo work is done with cells from a fourth patient: even though they are all t(8;21)+ it seems that they were selected to provide the most suitable results and/or

sufficient material to perform a particular experiment. This should be clearly stated throughout the MS.

Response: We apologise if our manuscript gives the impression that we had cherry-picked results which would indeed amount to scientific misconduct. t(8;21) samples do not come up every day and we worked with the material we had. This fact was one of the major reasons why we put in a big effort to finally create a PDX model for this type of AML, which, when published here, will be available for the scientific community. In Assi et al., we went through great length to show that omics data from t(8;21) AML cluster tightly together and we therefore felt justified to use different patients to address question addressing common mechanisms of t(8;21) AML biology. We simply cannot do every assay with every patient due to limited material. However, to satisfy the reviewer, we confirmed LSC-specific expression of VEGFA and IL5RA in the third patient, and a further patient (see also below and in response to reviewer 3) using single cell RNA-Seq all of which integrate well to show a common expression pattern, and which together with flow cytometry data of the VEGF and IL-5 receptors in the PDX from a fifth patient hopefully convinces them that our observations are real. We completely rewrote the first result chapter to accommodate the additional data and have added notes to explain why not all patients were used for certain assays.

Detailed comments:

1. Lines 115-118: The authors find expression of the AML-ETO transcript in 73 and 58% of cells “across all LSC and blast clusters confirming they were all AML or pre-leukemic”. Here the authors should dissect LSC and bulk. The illustration (Fig.1F) suggests a generally higher expression of the fusion in the LSC compartment than the bulk. In addition, there seems to be a continuum from LSC to bulk cells: is this also somehow correlating with the level of fusion mRNA? Could this be quantified on a single cell level? The reviewer however, does not understand the statement that fusion gene expression in a fraction of the cells suggests that the cells are “all AML or pre-leukemic”? Is AML-ETO not seen as a preleukemic initiation event in this type of AML? Why do some cells express low levels (or not at all) the fusion? Can they separate fusion + from fusion – cells and compare biologic behavior?

Response: We have investigated in more detail if there is a continuum of *RUNX1T1* expression from LSC to Blasts but the evidence for this is limited and we cannot draw any firm conclusions. Violin plots of the expression values across the clusters in the original version of the paper, ordered by pseudotime L-R are shown below to confirm this for the benefit of the reviewer (patient 1, left; patient 2, right).

The reviewer is correct that RUNX1::ETO is a preleukaemic initiation event, as such cells not expressing it would be expected to be contaminating healthy cells. However, scRNA-seq does have a high dropout rate, with only a limited number of genes detected per cell and so we cannot say with certainty whether expression is not detected due to technical artifact or true lack of expression. As we find the cells not expressing RUNX1::ETO distributed within all clusters rather than forming a specific subcluster, we can therefore infer that the lack of expression is due to dropout rather than contaminating healthy cells. We have revised this sentence to clarify this notion. However, please note that the addition of 2 more patients strongly improved our results but did not alter this conclusion.

2. To validate the nature of CD34+/CD38- cells as LSC, the authors compared the clonogenic activity of them with the CD34-/CD38+ bulk cells (Extended data figure 1B). Why was this done only with cells from patient #2? Did the bulk cells form clusters? The picture of a sole colony is not really informative: one would like to have a broader picture with e.g. type of colonies? It may be difficult if material is sparse, nevertheless the overall heterogeneity of the two patients studied implies a direct comparison.

Response: We understand that referees' concern, but primary t(8;21) AML is notorious for its poor colony formation ability, we and others have observed this behaviour (e.g. Griessinger et al. Exp Hematol 2018), and unfortunately none of the other patients nor the PDX formed colonies. We believe the ability of Patient #2 to form a limited number of colonies is due to a FLT3-ITD mutation. We have added a few words to the text to indicate this issue. The gating strategy has been previously validated with 3 other patients which carry FLT3-ITD mutations but not t(8;21), this unpublished data is shown below. LSC-derived colonies were similar in appearance to those shown in Supplementary figure 1B. We have also added pictures of 3 other colonies to Supplementary figure 1B from patient 2 to confirm that they were all similar in appearance to each other i.e. immature with no obvious terminally differentiated cells present.

3. Fig.1G shows a heatmap of hierarchically clustered gene expressed in LSC and bulk clusters from two patients: they marked IL5RA and VEGFA enriched in the LSC compartments of both of the patients. Where other genes significantly differentially expressed in LSC from both of the patients?

Response: numerous t(8;21) specific genes are enriched in the LSCs and can be identified on the heatmaps e.g. *POU4F1*, *PAX5* (mentioned in text), *TINAGL1*, *CNBD2*, *ROBO1* which may be of further interest to other researchers. In particular *PAX5* is interesting as we showed that it is the reason for the bi-phenotypic appearance of t(8;21) cells (Walter et al., 2010) and its expression in LSCs indicates that these cells still show signs of B-cell potential – which is another line of research altogether. In this manuscript we were interested in signalling, and no other genes associated with signalling processes were identified in this analysis. We therefore focussed only on VEGFA and IL5RA.

4. Fig.2A shows the cell cycle gene expression distribution of cells from both patients. For this reviewer the quantitative figure (extended Fig.2A) is more informative: could the authors at least give some numbers/percentage in the MS text?

Response: percentages have been added to the text and the quantitative figure for the four combined patients added to the main figure.

5. Fig.2C compares phosphorylation status (CYTOF?) of signaling molecules in CD34+ AML cells from two patients (#2 and #3). This is difficult to understand. First, why now another patient? If sufficient

viable material was not available from #1, why was all the epigenomic analysis done with this patient and NOT with patient 3? Or was this analysis also done but did not provide “suitable” data? In addition, the authors claim significant higher phosphorylation of CREB, JUN and JNK1/2 in bulk vs. LSC while e.g. STATs seem not changed: however the numbers in the right part of the panel suggest overall minor changes and in fact increased pSTATs in LSC than bulk. They need to present that in a clearer way that also “non-experts” can follow.

Response: See comments above. The data are indeed CYTOF data. The patient material used for each experiment was based on what was available at that time as these types of studies use millions of cells. No more material is left from patient 1 and CYTOF cannot therefore be carried out on this patient. In response to this comment and the comment from reviewer 3 we have now added patient 3 to the scRNA-seq analysis and due to low number of cells remaining and therefore retrieved from this patient, an additional new sample.

We apologise for the confusion in our presentation of this figure, the right panel is to display the levels of phosphorylated molecules in comparison to each other rather than between blasts and LSCs which is better represented by the fold change. As this is on a log-scale, differences do appear minimised. We have therefore replaced this figure with one showing only the levels in blasts (& healthy cells in response to reviewer 3).

6. Line 142: “...whether at LSC use...” is this sentence correct?

Response: Thank you for pointing this out. This was a typographical error and has been corrected.

7. Lines 155-157: The reviewer does not understand the statement that activating KIT mutations be insufficient to initiate LSC growth despite being found equally in LSC and blasts? Is this shown here? Previous studies showed that (viral) overexpression of activating KIT mutations cooperate with AML-ETO to induce a leukemic phenotype (<https://doi.org/10.1073/pnas.1019625108>; <https://doi.org/10.1182/blood-2011-02-338228>)? It can well be that some LSC have an activating KIT mutation and are quiescent, as following the stem cell hypothesis, only a minority of LSC are actively cycling? Could it be that the expression level of the fusion determines expression of putative growth regulators: it is interesting that IL5RA and VEGFA seems at least in part to correlate (Figs. 1F and 2D)? How does IL5RA and VEGFA correlate with the G1 phase of the cycle at a single cell level?

Response: For the benefit of the reviewer, we show here the gap in some of the aligned reads corresponding to the KIT mutation in both LSCs and Blasts. Due to the nature of sequencing data we cannot readily include all reads carrying the mutation in the manuscript.

On further inspection, we have found that KDR and KIT expression is in fact mutually exclusive - they share an enhancer as seen in HiC assays, and we also see this behaviour at the single cell level. In the KDR expressing LSCs, the KIT gene is not expressed so an activating mutation would not be present on these cells. This result is now shown and discussed in Figure 2F. We thank the reviewer for drawing our attention to this interesting mechanism of how LSCs may maintain quiescence in the presence of a KIT mutation, which is a mutation which very commonly co-occurs with t(8;21) AML.

IL5RA and VEGFA expressing cells do not show a significantly different distribution of cell cycle phase as compared to the rest of the LSCs presumably because this is a dynamic process (Patient 1, VEGFA 78% G1, IL5RA 82% G1, overall 84% G1; patient 2, VEGFA 55% G1, IL5RA 55% G1, overall 53% G1).

The responsiveness of cells to VEGF and IL-5 signalling is absolutely dependent on the presence of RUNX1::ETO. See our response to Referee 3.

8. Experiments shown in Fig.4 show nicely reduction of LSC by blocking IL5RA (Benralizumab) or VEGF (Bevacizumab) in a PDX model of t(8;21)+ AML. One wonders whether such a treatment also prolongates the latency until disease symptoms respectively survival (Kaplan-Meier plot)? Or would it at least translate into significantly altered of the PDX cells into a secondary host?

Response: Thank you. We agree this would be interesting, we would not expect much difference in survival without additional chemotherapy to remove the proliferating blast cells which we are unable to test currently. We attempted re-engraftment of cells from the first experiment but we did not observe human AML cells in peripheral blood by day 100 even in the vehicle controls, we believe there were too few cells remaining after other experiments. Unfortunately, as this is a 6 month+ experiment we cannot readily try again for this particular paper.

9. Why does the expression of dnFOS increases the clonogenic activity in the 2nd plate (replating)?

Response: AP-1 is required for myelopoiesis which is completely blocked after dnFOS induction (Obier et al.2016, Development). Note that at the second replating the number of colonies goes down due to the cell cycle and differentiation block, but we see a shift in clonogenic frequency as the cells are proportionally more immature as marked by the expression and binding of GATA2 seen in the liquid culture. We have added more explanations in the text to clarify this issue.

10. Fig7B shows differential gene expression in primary human normal CD34+ HSCP and CD34+ PDX cells upon inducible activation of dnFOS: how many and which genes were deregulated in both or more in the PDX cells?

Response: Very few genes de-regulated in the PDX were also de-regulated in the healthy cells: 1 upregulated gene was also up in healthy cells whilst 3 were down in the healthy cells, whilst 7 downregulated PDX genes were also down in healthy cells. We have added a heatmap to Supplementary Figure 7B to show this result and the genes are written in Supplementary Table 4.

It is difficult to understand that AP-1 seems required for the maintenance of the blast genetic program as initially the authors stressed expression and activation of IL5RA and VEGEF leading to AP-1 activation in LSCs? In addition, how does (as shown in the schematic 8F) activation of AP-1 in LSC will induce myeloid transcription factors in the blast cell compartment? Would that mean that AP-1 would shift the LSC towards the blast compartment: in other words, more IL5RA and VEGEF-induced AP-1, the smaller the LSC compartment? This is confusing

Response: We are aware that the regulatory circuit we describe here is complex. In our network diagram we have tried to explain how we envisage the different factors to control proliferation and self-renewal. We envisage RUNX1-ETO and AP-1 as part of a balance mechanism whereby RUNX1-ETO interferes with the cell cycle / differentiation and AP-1 drives it. RUNX1-ETO also drives expression of AP-1 which requires signalling to boost expression and allow binding to chromatin via post-translation modification etc. However, once AP-1 activity is established by hijacking ectopic signalling pathways, the circuit is compatible with both self-renewal and proliferation. We are not losing self-renewal but gaining growth of blasts due to signalling via AP-1.

To test this idea, we have measured the proportion of CD34+CD38- cells present after 6 days culture of patient 3 cells with or without VEGF and IL-5 to stimulate AP-1 activation, shown in the figure below for the reviewer. No significant difference was observed and so it appears that self-renewal is preserved but further experiments will be needed to understand how this is regulated going forward.

Reviewer #2

In this paper the authors show that patients with acute myeloid leukemia (AML) sub-type t(8;21) harbor leukemic stem cell (LSC) that aberrantly activate VEGF-A and IL-5 signaling pathways. Incubation of AMLs with VEGF-A and IL5 results in the expansion of LSCs. In a PDX model inhibition of IL5 and VEGF-A signaling diminishes the expansion of the LSCs. Employing ATAC-seq and ChIP analyses carried out on Kasumi cell line, they show that dysregulation of the oncoprotein RUNX1::ETO and activation of AP-1/GATA2 axis augments t(8;21) LSCs self-renewal and progression.

Response: This summary is largely correct but it is the balance between GATA2 (self-renewal) and AP-1 (progression) on which the signalling impacts together with RUNX1::ETO. We hope we have now clarified this by addressing other comments.

Comments:

Identifying the molecular pathways that are overactive in certain leukemias such as t(8;21) AMLs could lead to development of novel strategies to target leukemic stem cells (LSCs). The finding that t(8;21) in AML results in the aberrant overactivation of VEGF-A and IL5 could lead to designing new approaches to target LSC-driving leukemias.

Thank you

However, targeting VEGF-A and/or IL5 results in a modest response in leukemic progression. In addition, numerous previous have shown that inhibiting VEGF-A and its various receptors such as VEGFR2 or VEGFR1 only partially induce regression of leukemias.

Response: We agree that in trials e.g. HOVON-SAKK which we have referenced in the manuscript, inhibition of VEGFA showed only modest response. However, note that the HOVON-SAKK trial measured the response of a pan-AML cohort which included at most 2 t(8;21) patients, potentially none (genotypes were not reported). The mechanism we describe here is exquisitely t(8;21) specific (see Figure 2E and S3F). Our results provide further evidence that each AML sub-type has to be studied (and therapeutically targeted) individually.

Most importantly, the mechanism by which targeting VEGF-A and IL5 results in LSC regression or decreased self-renewal is unclear.

Response: This is of course a very interesting question, but in vivo we cannot distinguish between LSC loss and these cells changing fate i.e. becoming more differentiated once they are start to grow in response to cytokine signalling. The fact that *GATA2* is down-regulated in the blasts points to the second mechanism. We have some preliminary data which add to this idea (see below, LSCs begin to lose their gene signature) but to rigorously explain the molecular mechanism will be the subject of further research. This is a whole project in its own right.

Blast vs LSC fold change from PDX mice vs “blast” gene signature from patients

Blast vs LSC fold change from PDX mice vs “LSC” gene signature from patients

Figure legend: Gene set enrichment analysis comparing the gene expression fold change of Blasts/LSCs sorted from bone marrow of PDX-engrafted mice to the t(8;21)-specific LSC and Blast gene signatures defined within this paper.

Specific comments:

1) How does t(8;21) in AMLs and activation of RUNX1::ETO and AP-1/GATA2 results in induction of IL5 or VEGF-A. Are IL5 or VEGF-A induced in leukemic cells themselves or in the niche cells?

As we show in our single cell experiments, VEGFA/VEGFR are expressed in LSCs and so is the receptor for IL-5. These sections have been heavily rewritten so this should now be clear. The regulation of these genes and which transcription factors and cis-regulatory elements activate them is shown in Figure 8, gene regulation is controlled by the combinatorial binding of various factors but GATA2 binding which is repressed by AP-1 is the key to the LSC-specific expression. Our network diagram describes the activation of VEGFA and *IL5RA* in fine detail. IL-5 was not detected in any leukaemic cells. RUNX1::ETO ultimately controls subtype specific expression of these factors (Figure 8A) and new Figure 8B (see also response to reviewer 3).

2) Which niche cells are presumably activated by LSCs, and how they are induced to produced to release VEGF-A or IL5?

Response: Answering this question is way out of the scope of this paper. It sounds like a simple question to answer, but it is not. The interaction of t(8;21) LSC and blast cells with the leukemic niche and how the niche regulates gene expression is of course a really burning question which can only partially be answered in mice and here it is unclear whether the answers are relevant for humans. It needs to be done properly. What we ideally need here is a human niche model or histology from human bone marrow biopsies from AML patients using a trephine (essentially a drill which preserves bone structure). We do not have such samples from our patients, only diagnostic samples from people who are already very stressed. We plan, however, to work with people putting together human niche models – a line of research that is still in its infancy. Preliminary gene expression analysis from human mesenchymal stromal cells co-cultured with t(8;21) AML (shown below for the reviewer) suggests the AML does not induce them to express more VEGFA, again IL-5 was not detected so this must be coming from other cells in the bone marrow which is a very heterogeneous population of cells.

Figure legend: UMAP of scRNA data from human mesenchymal stromal cells alone (green), co-cultured with a t(8;21) AML patient (blue) or non-t(8;21) AML patient (orange) on the left, with VEGFA expression projected on with green-yellow indicating highest expression (right).

3) Which receptors for VEGF-A (VEGFR1, R2 or R3) and IL5 are expressed on LSCs? Is the expression of these receptors dependent on the factors released specifically by t(8,21) and RUNX1::ETO or AP-1/GATA2 axis?

Response: Both subunits of the IL5 receptor are expressed, but only the alpha unit (IL5RA) was specific to t(8;21) LSCs. We could detect VEGFR2 (KDR) by RNA-seq and flow cytometry as described already in the manuscript. VEGFR1 (FLT1) was also expressed in LSCs but this also seems to be expressed in several other AML subtypes, VEGFR3 (FLT4) was expressed in both blasts and LSCs and seems to be specific to t(8;21) and inv(16) AML but to our knowledge is not canonically activated by VEGFA. Both VEGFA and KDR are responsive to RUNX1-ETO knockdown (shown in Figure 8 and Issa et al. 2023 Leukemia), see also response to reviewer 3. As mentioned in response to point 1, combinatorial binding regulates expression of these genes.

4) Why for the ATAC-seq and ChIP analyses the authors have elected to use Kasumi cell line rather than primary leukemic cells that harbor t(8,21) translocation.

Response: See our answer to reviewer 1. The Kasumi-1 cell line is a widely used and well established model for t(8;21). In fact, our lab has shown that the chromatin landscape and RUNX1::ETO binding pattern of these cells closely resembles that of primary cells (Ptasinska et al., 2012). We cannot do every assay with patient cells as limited numbers of cells were available from primary patient samples with t(8;21). We made a big effort to conduct experiments on primary cells where possible i.e. in Figures 1-4. For the experiments with dnFOS, the PDX was utilised but following lentiviral transduction only enough cells for RNA-seq were retrieved after cell sorting as the transduction frequency was low. ATAC-seq was attempted but data quality from such low cell numbers was too poor to use.

5) The authors should consider providing histological analyses of the leukemic bone marrow after targeting of IL5 and VEGF-A.

Response: See our answer above with regards to the caveats of such experiments. We have of course considered histology but decided that adding such data as well would make the paper even more ungainly. Now that we have the PDX system, we are in a fantastic position to study AML-cell niche interactions in fine molecular detail in different model systems without worrying about patient heterogeneity - in a different study, after we published this one.

Reviewer #3

In this manuscript, Sophie G Kellaway et al., by using a multidisciplinary approach, tried to dissect regrowing of LSCs, a process commonly associated with relapse induction after therapy. Specifically, the authors by using the t(8;21) AML sub-type as a model of investigation (cells, cell lines and patient-derived xenograft model (PDXs)) claim that VEGF and IL-5 signalling pathways are aberrantly activated in LSCs and are required for LSCs growth. Importantly the authors define that both pathways act via

regulatory circuits formed by RUNX1-ETO and an AP-1/GATA2 axis giving rise to the LSCs to re-enter the cell cycle while preserving self-renewal capacity.

Overall, the manuscript draft is detailed and readable representing a potential contribution to the field with an appreciable effort in identifying molecular patterns driving LSCs growth. However, some gaps in the manuscript prevent its publication in the current form and these concerns need to be addressed by the authors.

Response: We thank the reviewer for their astute and correct summary and positive comments.

Major comments

1- The authors identified an LSC signature by applying single-cell (sc) approaches in only two patients. Although the analysis seems to be well performed, the number of patients used is a limitation since it cannot represent a “common” line in LSCs and is undoubtedly not enough for building the roadmap of the paper and defining identified genes as involved in the LSC maintenance. Thus, the authors should implement this data by using more patients (for example they used pt 3 and pt 4 in the paper, why do not include them at least for the most important and informative sc analysis?). Additionally, I found that expression levels of VEGF and IL5RA are quite variable in the two patients, and the correlation between RNA-seq and sc for KDR appears weak suggesting (also in this case) that more samples might be needed and likely of help.

Response: We have conducted scRNA on patient 3, and due to the low number of cells retrieved from this patient, a further new patient sample. No material is left from the diagnostic sample for patient 4 (now relabelled patient 5). The scRNA data from all of these patients integrate well and confirm our results, with expression of *IL5RA* and *VEGFA* confined to the LSCs of all 4, and *KDR* expression in the LSCs of the two samples with the most cells sequenced.

We would like to note that in Assi et al., we went through great length to show that omics data from t(8;21) AML cluster tightly together and we therefore felt justified to use different patients to address question addressing common mechanisms of t(8;21) AML biology, however the data provided here is now much stronger. See also response to reviewer 1.

2- To corroborate their findings, authors could compare results obtained from experiments performed in Fig 2 with “normal samples” (if possible) and analyze gene expression in normal blasts, HSC, and leukemia.

Response:

1. The CyTOF panel was initially optimised using healthy bone marrow and the mean ion counts from this have been added to Figure 2C, JUN and JNK antibodies were not tested on these samples and unfortunately no further material is available so this has been clearly indicated on the figure. Fold changes for mature/immature healthy cells are not shown as the overall level of these phospho-proteins were very low and as such the fold changes were not meaningful. This shows nicely the increased signalling present in AML and we thank the reviewer for their suggestion.

2. scRNA data from healthy cells were downloaded and analysed by the same method as the AML samples, this data has been added to Figure 2B and discussed in the text: LSC-specific terms were enriched in HSCs, blast-specific terms were not highly enriched in healthy cells, “translation” was detected in myeloid progenitors but not HSCs confirming our hypothesis.

3- To deeper confirm the involvement of VEGF and IL5 signalling on t8;21 leukemic cells the authors should carry out additional phenotypic analysis, such as CCK8 assay, cell cycle analysis, or competition assay (etc).

Response: We have carried out a G₀/G₁ cell cycle analysis which nicely shows that both the VEGF and IL-5 inhibitors revert SKNO-1 cells to G₀, effectively pushing them out of cell cycle. We have added this data to Figure 3D and thank the reviewer for this suggestion.

4- According to Fig 4J and F, the results displayed are not significant, can the authors address this point?

Response: We have repeated this experiment to confirm what we saw and found a similar trend, with additional mice reducing the p-values when significance was tested. Fig4F + bevacizumab shows p = 0.0035 with benralizumab p = 0.0997, down from p = 0.15. Due to the small effect size (targeting ~2% of cells, with NSG mice only expressing a small amount of IL-5) a power calculation suggests we would need between 13 and 23 mice per group to detect a significant difference with benralizumab, we have added a sentence to explain this finding. The differences seen in Figure 4J are now p<0.05 with the additional mice.

5- It is unclear why the authors chose Kasumi-1 as a cell model to investigate molecularly the role of AP1 since this cell line does not harbor IL5R. Can the authors better clarify this point? (Fig 6)

Response: We have explained the use of this model above, this is the best-characterised model of t(8;21) AML. Furthermore, for genome-wide studies we have previously produced promoter capture Hi-C data for this cell line which means we can correctly assign cis-regulatory elements to their genes which is crucial for transcription factor binding analysis. Kasumi-1 cell indeed carry a deletion of the IL5RA locus, but it is still a good model to look at the role of AP-1 in mediating the effect of VEGFA, as well as the global phenomena of transcription factor binding. We have also constructed SKNO-1 expressing dnFOS which we show are IL-5 responsive when AP-1 is not blocked in Figure 5.

6- Based on the results, on one hand, it seems that targeting AP1 contributes to affecting leukemic phenotype, however, it also induces LSC proliferation via GATA binding. Such a story has only been persecuted molecularly without a phenotypic readout (if present is very poor). Therefore, could the author asses phenotypically that targeting AP-1 induce a shift from LSC to blast cells? Additionally, authors should better explore (and also comment on) how targeting AP1 implies leukemic cell growth and at the same time induces LSC self-renewal and proliferation.

Response: We apologise for the misunderstanding, we obviously explained this really badly. Blocking AP-1 binding activity using dnFOS or cytokine inhibitors does not send LSCs into proliferation, activating AP-1 via cytokine signalling does. From our previous work (Martinez-Soria et al., 2018) we know that dnFOS treatment causes a cell cycle block in t(8;21) cells, we know that AP-1 binds to cell cycle genes which are also bound by RUNX1-ETO and RUNX1, we know that these genes are down-regulated after dnFOS expression. dnFOS also completely blocks tumourigenesis in vivo (Assi et al., 2029). As explained in our response to Reviewer 1, signalling through AP-1 is required to kick-start LSC growth which then turn into blast cells with some cells retaining self-renewal potential. We appreciate this is a complex mechanism due the shifting balance of combinatorial transcription factor binding which is why we

have summarised this in the model in Figure 8. Essentially, when AP-1 is activated by signalling the transcription factor binding landscape shifts from a GATA2 dominated signature which codes for self-renewal to a RUNX1/PU.1/CEBPA signature which codes for proliferation and an attempt to differentiate to the myeloid lineage, which is blocked by RUNX1::ETO. This is a dynamic process in our cultures and as such blocking AP-1 or blocking signalling increases the relative proportion of cells which can only self-renew, not proliferate. We have amended the text describing Figure 8G to explain this better.

7- Fig 8. Can authors corroborate and strengthen these findings by performing experiments by knocking down RUNX1-ETO following IL5 and VEGF stimulation?

Response: We have combined RUNX1-ETO knockdown with VEGF stimulation in the Kasumi-1 cell line with pre-treatment with either RUNX1-ETO or VEGF, this data has been added to Figure 8B. No response to VEGF stimulation was seen with RUNX1-ETO knockdown confirming that RUNX1-ETO expression is required for this response, pre-stimulation with VEGF did not rescue the effects of knockdown on growth. We do not have a stable shRUNX1-ETO SKNO-1 line and so did not do the same experiment with IL-5. See also our response to Reviewer 1.

Minor comments

To confirm the specificity of the pathways (IL5 and VEGF) could authors analyze the effects of these cytokines also in other leukemic models (by using cell lines)?

Response: We thank the referee for this good suggestion. This is a valuable control which we have now added to Supplementary Figure 3F – no effect is seen in non-t(8;21) models confirming the specificity.

Figure 4D shows sampling at different time points of blood samples, however, it is not clear how these samples have been used in the results readout.

As blood was not serially sampled for all mice in the repeat experiment we have removed this figure and now only show the % engraftment relative to controls at the harvest time point.

Stdev needs to be added in FigS1C.

Response: This graph shows 1 biological replicate, with technical replicates only and so error bars are not appropriate here. We have updated the figure legend to clarify this notion.

We hope that the above-listed changes now make our paper suitable for publication in Nature Communications.

Yours sincerely

Constanze Bonifer and Sophie Kellaway

on behalf of all other authors.

REVIEWERS' COMMENTS

Reviewer #1 (Remarks to the Author):

The authors have really tried to address all my comments: as noted by them, "it is work in progress". Nevertheless this overall interesting MS clearly improved. However, based on the fact that the data shown here seems rather specific for t(8;21)+ AML, should this not also be reflected in the title of the MS?

Reviewer #2 (Remarks to the Author):

The authors have performed additional experiments and have addressed the majority of my concerns. I just have a few concerns remaining.

1) The authors state that Kasumi cell line do not have IL5 receptor and yet they could be targeted by IL5 receptor inhibitors. Could they provide explanation for this inconsistency?

2) Could they show that indeed KDR is functional on the primary leukemic cells and upon activation with VEGF-A can undergo phosphorylation or it performs primarily as a low activity kinase?

3) Does it matter which isoform of VEGFA was used in these studies? VEGFA121 or VEGFA165. Is neuropilin play a role in this signaling?

4) How does KDR activation regulate the pro-leukemic function? Through proliferation or mainly survival? How about interaction with niche cells, such as adhesion or migration?

5) Similarly, how does IL5 support the pro-leukemic function through Jak/Stat pathway activation? Survival or augmented interaction with the niche cells.

6) Does small molecule tyrosine kinase inhibitors targeting KDR or Jak/stat block leukemic stem cells as well?

Reviewer #3 (Remarks to the Author):

The authors have replied with experiments to the majority of the comments and the manuscript is now more clear and definitely more focused. Some of the concerns have been answered with citations of the literature or with clear technical limitations which are understandable

RESPONSE TO REVIEWERS' COMMENTS

REVIEWER #1

The authors have really tried to address all my comments: as noted by them, "it is work in progress". Nevertheless this overall interesting MS clearly improved. However, based on the fact that the data shown here seems rather specific for t(8;21)+ AML, should this not also be reflected in the title of the MS?

Response: We thank the reviewer for their positive evaluation. However, we respectfully ask to maintain the title generic. Our proof of principle experiments required a multitude of experiments to resolve a mechanism and we had to concentrate on one AML sub-type as a paradigm for the hijacking of ectopic signalling pathways. Our work shows that the expression of the VEGF/KDR and IL5R ectopic signalling axis is indeed t(t;21) AML sub-type specific. However, as outlined on p14 in the discussion we believe that ectopic activation of signalling processes is a phenomenon that is employed by multiple AML sub-types as shown by our own work analysing the C/EBP mutant sub-type and others, with signalling-responsive AP-1 overexpression a core component of all studied AML sub-types (Supplementary Figure 8A, Assi et al 2019). We have now added the C/EBP mutated AML expression data into Supplementary Figure 8I-J to illustrate this point. We also changed the title into "**Leukemic stem cells activate lineage inappropriate signalling pathways to promote their growth**" which more precisely describes what is happening to reflect this.

REVIEWER #2

The authors have performed additional experiments and have addressed the majority of my concerns. I just have a few concerns remaining.

Response: We thank the reviewer for their positive evaluation. However, as reviewer 1 has already stated, this is a work in progress and there is only so much we can put in one paper.

1) The authors state that Kasumi cell line do not have IL5 receptor and yet they could be targeted by IL5 receptor inhibitors. Could they provide explanation for this inconsistency?

Response: We apologise for the misunderstanding. Experiments with benralizumab (IL-5 receptor inhibitor) were only performed with SKNO-1 and our PDX model which does not have the deletion that took out *IL5R* in Kasumi-1 cells, whilst experiments with bevacizumab (VEGFA inhibitor) were additionally performed on Kasumi-1. All figures have now been clearly labelled with the cell line used, and separated out and colour coded blue for VEGF/bevacizumab and orange for IL-5/benralizumab throughout the manuscript as we realise the very similar names of the inhibitors is a source of confusion.

2) Could they show that indeed KDR is functional on the primary leukemic cells and upon

activation with VEGF-A can undergo phosphorylation or it performs primarily as a low activity kinase?

Response: This experiment is not currently possible in the leukaemic stem cells as the numbers of cells expressing KDR are way too low. However, we are working on ways to study receptor activation and downstream mechanisms in LSCs. For the benefit of the reviewer, we attach preliminary data from a Full Moon BioSystems phosphorylation array with Kasumi-1 cells with knockdown of RUNX1-ETO (siRE) or control (siMM). Here we see that in basal growth conditions (i.e. with autocrine VEGFA/KDR stimulation) KDR is phosphorylated, and both phosphorylated and total KDR were lost with RUNX1-ETO knockdown. This assay is only semi-quantitative in our experience and difficult to normalise. Therefore, more work will be required to confirm the details of this finding, and test phosphorylation in an LSC-specific manner.

3-1) Does it matter which isoform of VEGFA was used in these studies? VEGFA121 or VEGFA165.

Response: VEGFA165 is the important isoform here, see below genome browser screenshot of bulk RNA-seq data (Assi et al. 2019 Nature Genetics), and it is recombinant VEGFA165 we used to stimulate growth as detailed in the methods section.

3-2) Is neuropilin play a role in this signaling?

Response: This is a very interesting question. Our data show that the VEGF receptor co-factor Neuropilin-1 is not expressed in t(8;21) AML but neuropilin-2 seems to show increased expression in the LSC-blast transition cells. Therefore, it may be the case that after quiescent cells receive the initial signal, expression is activated to boost signalling after the initial stimulation in LSCs in a process parallel to other immune signalling pathways. This is an interesting idea which we will certainly pursue going forward but again this would be a whole additional study. We have added this data to Supplementary Figure 2G-H and discussed it in the text and thank the reviewer for this suggestion.

4) How does KDR activation regulate the pro-leukemic function? Through proliferation or mainly survival? How about interaction with niche cells, such as adhesion or migration?

5) Similarly, how does IL5 support the pro-leukemic function through Jak/Stat pathway activation? Survival or augmented interaction with the niche cells.

Response: For both VEGF and IL-5 signalling we observed only growth effects, the inhibitors did not impact on survival/apoptosis. The idea of interaction with the niche vs migration is interesting, and something we had considered as we had noted that the expression of *RHOC* is largely LSC-specific. To this end we performed preliminary trans-well migration assays with dnFOS expressing Kasumi-1 cells. The observed fold change suggested a 10-20% reduction in the capacity for migration with dnFOS i.e. with signalling blocked and LSCs stalled, however the figure below shows that results were variable, effect size small and statistical significance was low.

Figure legend: Number of Kasumi-1 cells migrating through a transwell insert towards mesenchymal stem cells (MSCs) or SDF-1 α (left), and expressed as a fold change of plus dnFOS/minus dnFOS calculated per experiment (right).

6) Does small molecule tyrosine kinase inhibitors targeting KDR or Jak/stat block leukemic stem cells as well?

Response: Inhibition of STAT signalling causes apoptosis in Kasumi-1 (Redell et al. 2011 Blood, Ray et al. 2013 Blood) as well as disrupting healthy blood cell growth and viability hence a lack of clinical use. A VEGFR2 kinase inhibitor CAS 15966-93-5 has been previously shown to block growth preferentially of Kasumi-1 although LSCs were not studied (Hiramatsu et al. 2005 Leukemia and Lymphoma) suggesting that the receptor can indeed be targeted for the specific phenomenon we observe. From the point of view of drug repurposing we focused our efforts here on bevacizumab but new drugs being worked on for the receptor will likely be of use in the future.

REVIEWER #3

The authors have replied with experiments to the majority of the comments and the manuscript is now more clear and definitely more focused. Some of the concerns have been answered with citations of the literature or with clear technical limitations which are understandable.

Response: We thank the reviewer for their positive evaluation.

We hope that our revision has answered all remaining questions, and the paper is now suitable for publication in Nature Communications.

Constanze Bonifer and Sophie Kellaway

On behalf of other authors.